# FastGrasp: Learning-based Whole-body Control method for Fast Dexterous Grasping with Mobile Manipulators

## Abstract

Fast grasping is critical for mobile robots in logistics, manufacturing, and service applications. Existing methods face fundamental challenges in impact stabilization under high-speed motion, real-time whole-body coordination, and generalization across diverse objects and scenarios, limited by fixed bases, simple grippers, or slow tactile response capabilities. We propose **FastGrasp**, a learning-based framework that integrates grasp guidance, whole-body control, and tactile feedback for mobile fast grasping. Our two-stage reinforcement learning strategy first generates diverse grasp candidates via conditional variational autoencoder conditioned on object point clouds, then executes coordinated movements of mobile base, arm, and hand guided by optimal grasp selection. Tactile sensing enables real-time grasp adjustments to handle impact effects and object variations. Extensive experiments demonstrate superior grasping performance in both simulation and real-world scenarios, achieving robust manipulation across diverse object geometries through effective sim-to-real transfer.

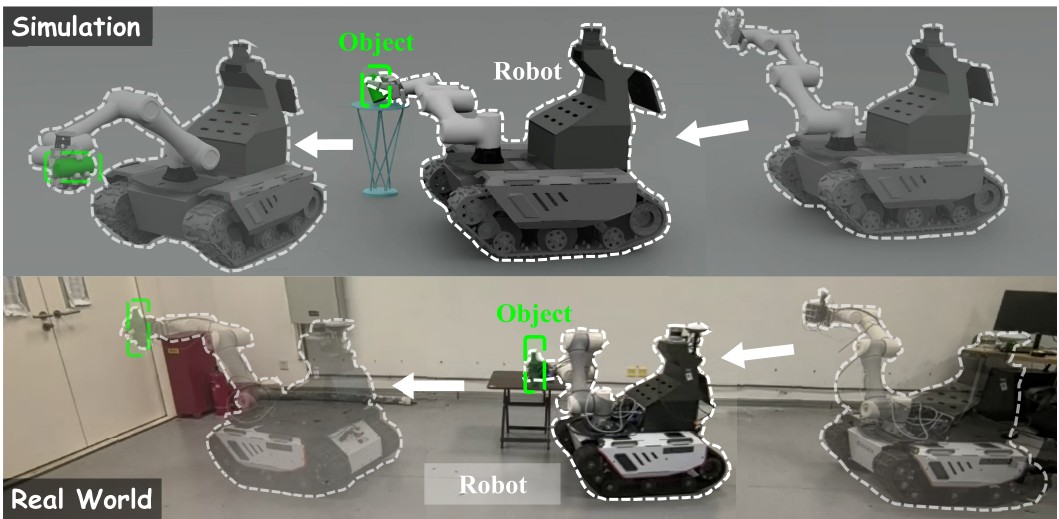

Figure 1: **FastGrasp** demonstration in simulation and real-world scenarios.

## 1 INTRODUCTION

Fast grasping capability refers to the comprehensive ability of dexterous mobile robots to accurately grasp target objects during high-speed motion, representing a critical technology for precise manipulation and efficient operations. This capability holds significant application value across multiple domains: in logistics and warehousing, it can substantially enhance sorting efficiency; in domestic service, it improves user experience; and in industrial manufacturing, it increases production throughput. Mastering fast grasping represents not only technological advancement but a pivotal breakthrough in the transition from static operations to dynamic interaction paradigms in robotics.

However, existing research on fast grasping exhibits significant limitations in multiple dimensions. Static dexterous manipulation approaches (Wei et al., 2024), (Zhang et al., 2025b), (Zhang et al., 2025a) achieve remarkable precision through sophisticated grasp synthesis methods but are fundamentally constrained by fixed base limitations, rendering them incapable of handling objects beyond their predefined workspace. Conversely, mobile manipulation systems (Burgess-Limerick et al., 2023), (Haviland et al., 2022), (Burgess-Limerick et al., 2024) provide enhanced workspace coverage but typically employ simple grippers, severely limiting their dexterous manipulation capabilities. Additionally, while tactile-enhanced systems (Chebotar et al., 2014), (Lee et al., 2024) show promise in compensating for visual limitations, they either require substantial computational resources or lack the rapid response capabilities necessary for high-speed dynamic grasping scenarios.

These limitations give rise to several fundamental challenges in this task of fast grasp with mobile base and dexterous hand: (i) **stable grasping of objects under high-speed motion**: the significant velocity differential between a high-speed robot and a target object induces substantial impact effects during grasping, often resulting in object rebound, rotation, or slippage, and the extremely narrow contact time window imposes stringent requirements on the real-time performance of the perception-control system, where even minor temporal errors or pose inaccuracies will lead to grasp failure; (ii) **real-time whole-body coordinated motion planning**: unlike static dexterous systems that achieve high precision in fixed workspaces or mobile systems that use simple grippers, fast dexterous grasping requires seamless coordination between high-speed base motion and multi-fingered manipulation, performing precise grasping operations during high-speed movement while ensuring whole-body motion coordination. This demands the synergistic integration of real-time perception, dynamic motion planning, and adaptive control under strict temporal constraints; (iii) **grasping objects with diverse shapes and sizes**: objects of varying geometries and varying poses in diverse environments demand a control strategy with high generalization capability and robustness.

To address these challenges, we propose **FastGrasp**, a learning-based framework that integrates whole-body control, grasp guidance, and tactile feedback for mobile fast grasping. Our approach tackles the stability issues under high-speed motion by employing a two-stage policy that first generates diverse grasp candidates and then guides the execution to minimize impact effects. For real-time whole-body coordination, we develop a unified reinforcement learning framework that simultaneously controls the mobile base, arm, and hand, ensuring seamless integration of navigation and manipulation. To handle objects with diverse shapes and sizes, we incorporate tactile feedback that enables rapid grasp adjustments and enhances generalization across varying object geometries and environmental conditions. Our policy is trained in simulation and deployed on real robots through sim-to-real transfer. Our key contributions are summarized as follows:

1. We propose a learning-based approach that first solve the problem of simultaneously coordinates the mobile base, arm, and dexterous hand movements for high-speed grasping tasks, which is significant for real-world robotic applications.

2. We propose to employ a pretrained grasp pose generator to deliver high-quality grasp candidates, thereby guiding the learning of dynamic grasping policies and enhancing grasp stability during rapid approach and contact.

3. We integrate tactile sensing to enable real-time grasp adjustments, enhancing robustness across diverse object geometries.

4. We demonstrate effective ~~zero-shot~~ transfer from simulation to real robots, achieving superior performance in both simulated and real-world experiments, validating the framework's practical applicability and effectiveness.

## 2 RELATED WORK

### 2.1 DEXTEROUS MANIPULATION

Dexterous manipulation has received significant attention in robotics ((Liu et al., 2024), (Ze et al., 2024), (Wei et al., 2024), (Wang et al., 2025), (Dasari et al., 2022), (Zhang et al., 2025a), (Zhang et al., 2024), (Zhong et al., 2025a), (Zhu et al., 2025),(Zhong et al., 2025b)), which enables robots to be applied in a wide range of fields((Bicchi, 2002), (Okamura et al., 2000), (Ma & Dollar, 2011), (Nagabandi et al., 2020)), with dexterous hands (Shadow Robot Company, 2023) offering superior

flexibility compared to traditional grippers. However, their high-dimensional control spaces present significant challenges for learning-based policy training. Recent approaches address these challenges through various strategies: (Xu et al., 2023) introduces goal-conditioned policies for complex execution, (Wei et al., 2024) predicts feasible grasps with strong generalization, and (Zhang et al., 2025b) enhances robustness through hand-centric shape representations. However, these systems remain fundamentally constrained by fixed base configurations, limiting their operational workspace. Our approach addresses this limitation by integrating mobile bases with dexterous hands, significantly enhancing flexibility and environmental adaptability for grasping tasks.

## 2.2 MOBILE MANIPULATOR BASED ON WHOLE-BODY CONTROL

A simple method of whole-body control is the sequential execution of base and arm movements ((Quigley et al., 2009), (Wang et al., 2009), (Li et al., 2017)). Building on dexterous manipulation advances, recent works have explored integrating mobility through whole-body control strategies. Existing approaches can be categorized into optimization-based methods (Shan et al., 2004), (Colombo et al., 2019), (Zimmermann et al., 2021), (Burgess-Limerick et al., 2023; 2024), (Haviland et al., 2022) that formulate control as quadratic programming problems with theoretical guarantees but struggle with complex dynamics, and learning-based methods (Bajracharya et al., 2024), (Hu et al., 2023), (Feng et al., 2025), (Zhou et al., 2025), (Zhang et al., 2024) that employ reinforcement and imitation learning for complex tasks. However, these approaches share key limitations: they primarily use simple two-finger grippers rather than dexterous hands and inadequately address real-time coordination for high-speed scenarios. While (Zhang et al., 2024) represents progress toward high-speed manipulation with a two-stage framework for rapid object catching, it lacks the grasp precision needed for diverse objects. Building upon (Zhang et al., 2024) and (Xu et al., 2023), this paper presents a grasp-guided approach for whole-body control policy learning in fast grasping tasks, enhancing both coordination and efficiency.

## 2.3 TACTILE-BASED MANIPULATION

While grasp guidance enables effective policy learning, real-time adaptation during execution is crucial for high-speed scenarios. Tactile sensing compensates for visual limitations ((Chebotar et al., 2014), (Lee et al., 2021; 2024)) and enables robust grasping of dynamically moving objects (Liu et al., 2017). Tactile sensing can enhance the ability to interact with deformable objects (Kaboli et al., 2016), enable robots to have performance capabilities that are closer to those of humans ((Hogan et al., 2020), (Pirozzi & Natale, 2018), (Delgado et al., 2015)). Existing methods employ either computationally intensive vision-based tactile approaches using high-resolution imagery ((Wu et al., 2025), (Funabashi et al., 2020), (Chebotar et al., 2014), (Murali et al., 2022)), or more efficient pressure-based representations ((Lee et al., 2024), (Zhang et al., 2025b), (Yang et al., 2024)). However, adapting these for rapid response remains challenging. Our system addresses this by using simplified binary contact feedback for efficient policy inference and reduced sim-to-real gap, enabling real-time grasp adjustments during high-speed motion while capturing key physical interactions critical for stabilization.

## 3 SYSTEM SETUP

**Real-world Setup:** As shown in Fig. 2, the robot consists of an Agilex Bunker Mini mobile base, a 6-DOF Dobot CR5 arm, and a 13-DOF LeapHand. We equipped 9 resistive thin-film pressure sensors on each finger and palm to obtain binary tactile information, and installed a RealSense D435i camera at the end-effector of the robotic arm to provide ego-view RGB-D visual information. For onboard computation, we use an NVIDIA Jetson AGX Orin. All components of our robot are powered by the 48V power interface from the Agilex Bunker Mini. We use ROS to manage the various components of the mobile manipulator. The RGB-D camera captures the objects' point cloud, and the policy performs inference and publishes control commands at 25 Hz, maintaining the same frequency as in simulation to ensure consistency during real-world deployment.

**Simulation Setup:** We choose Isaac Sim (NVIDIA Corporation, 2022) as our simulation environment. In the initial state, as shown in Fig. 2, the robot remains stationary and the object is placed on a table at least 2.0 meters in front of it to ensure sufficient distance for base acceleration. Each

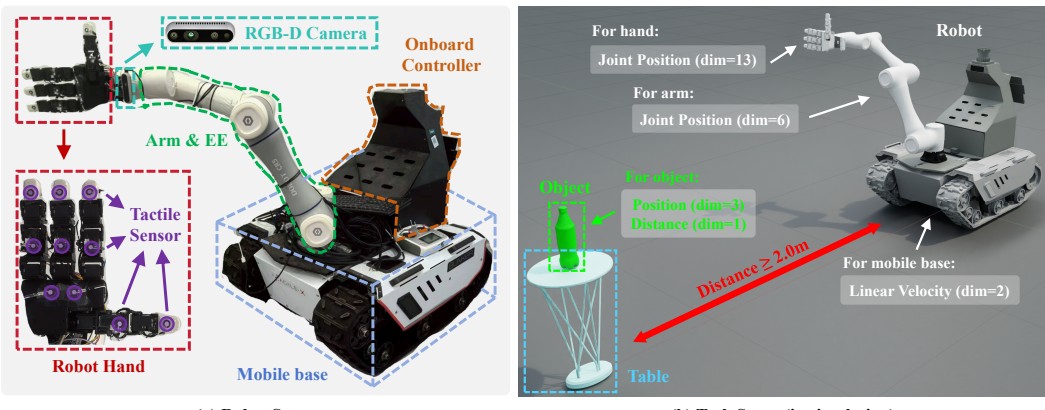

(a) Robot Setup  (b) Task Setup (in simulation)

Figure 2: System Setup

episode terminates and resets upon unintended collision or object dropping. During reset, the robot returns to its initial state while the object is randomly relocated to arbitrary positions within the region illustrated in Fig. 2 and orientation on the table in front of the robot. To enhance data diversity, each parallel environment is initialized with a distinct object and random table height.

**Robot Setup:** We replicate the real robot structure in simulation using URDF models, ensuring identical geometric and kinematic properties between simulated and physical robots: **(i) Mobile base**: Controlled via forward and yaw velocity commands, with maximum forward velocity of 1.3 m/s and maximum yaw velocity of 1.0 rad/s, capable of accelerating from standstill to maximum forward speed in 0.7 seconds. **(ii) Arm**: Controlled via joint position commands with maximum joint movement speed of 100°/s. For safety considerations and to prevent uncontrolled movements during high-speed motion, we fix the fourth joint to remain horizontal. **(iii) Hand**: Controlled via joint position commands with maximum joint velocity of 90°/s.

## 4 LEARNING FAST DEXTROUS GRASPING POLICY WITH TACTILE SENSOR

### 4.1 OVERVIEW

Our goal is to enable whole-body robots to rapidly approach and grasp objects beyond their initial operational range while maintaining base velocity, then quickly retract from the table while ensuring object stability. Inspired by (Xu et al., 2023), our FastGrasp framework consists of two stages.

First, grasp guidance generation: To address generalizable grasping and reduce whole-body coordination complexity, we pre-train a conditional variational autoencoder (CVAE)-based generator to produce diverse grasp proposals from object point clouds (Sec. 4.2). The optimal grasp candidate is then selected by maximizing hand-to-object coverage to guide the RL policy toward stable and executable grasps (Sec. 4.3). Second, policy learning with tactile feedback: We train the policy using guidance information and real-time robot states as inputs to generate control commands for fast grasping execution. To address grasp failures caused by inertial forces during high-speed motion, we integrate binary tactile feedback to monitor physical interactions between the object and hand (Sec. 4.4). The complete framework is illustrated in Fig. 3.

### 4.2 GRASP PROPOSAL GENERATION

In this section our goal is to generate diverse grasp candidates using object point clouds. Inspired by (Mayer et al., 2022), we employ a conditional variational autoencoder (CVAE) (Sohn et al., 2015) for its rapid sampling and diverse output generation. As shown in Fig.3, given a set of latent samples $\mathbf{z}$, the point cloud feature $\mathbf{F}$ encoded by PointNet (Qi et al., 2017), and the successful grasp $\mathbf{g}$ conditioned on the $\mathbf{F}$, the model enables controllable generation by incorporating conditional information. Its core objective is to maximize the following evidence lower bound:

$$\mathbf{L}_{\text{CVAE}}(\theta, \phi; \mathbf{g}, \mathbf{F}, \mathbf{z}) = \mathbb{E}_{q_\phi(\mathbf{z}|\mathbf{g}, \mathbf{F})}\left[\log p_\theta(\mathbf{g} \mid \mathbf{z}, \mathbf{F})\right] - D_{\text{KL}}\left(q_\phi(\mathbf{z} \mid \mathbf{g}, \mathbf{F}) \| p(\mathbf{z})\right) \quad (1)$$

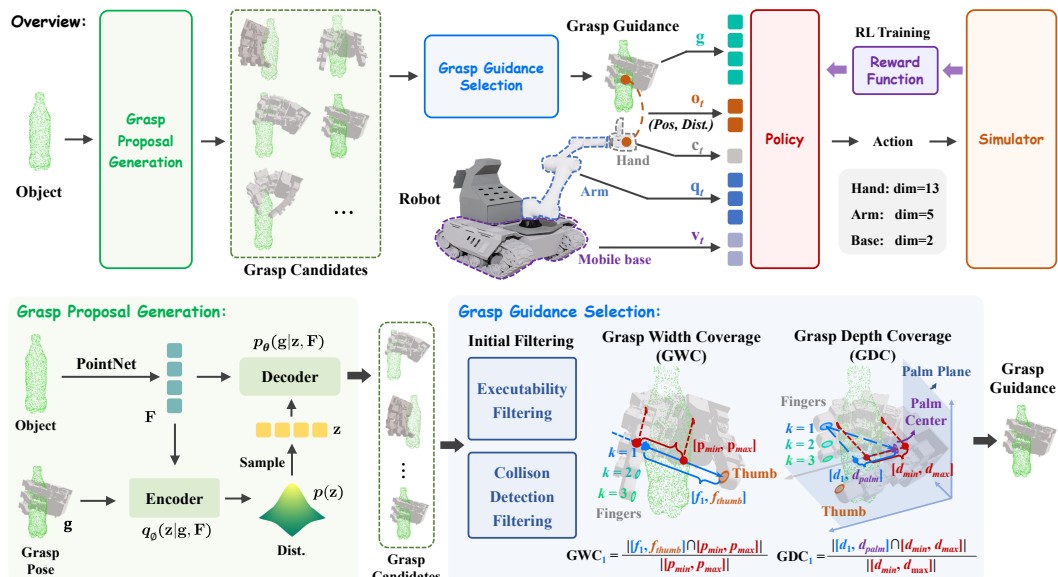

Figure 3: Overview of FastGrasp framework

where the encoder $q_\phi(\mathbf{z} \mid \mathbf{g}, \mathbf{F})$ infers the posterior distribution of latent variable $\mathbf{z}$ based on object point cloud feature $\mathbf{F}$ and grasp configuration $\mathbf{g}$, and the decoder $p_\theta(\mathbf{g} \mid \mathbf{z}, \mathbf{F})$ generates grasp poses using a latent code $\mathbf{z}$ sampled from a prior distribution $p(\mathbf{z}) = \mathcal{N}(\mathbf{0}, \mathbf{I})$ and the point cloud feature $\mathbf{F}$. By minimizing KL divergence to regularize latent space while maximizing likelihood of generated grasps during training, the CVAE learns to produce diverse and stable grasp candidates from input.

During inference, the decoder receives as input the concatenated representation of the point cloud feature $\mathbf{F}$ and a latent variable $\mathbf{z}$ randomly sampled from the Gaussian distribution. The grasp pose is subsequently reconstructed from this combined representation. To ensure sufficient grasp diversity, we generate a collection of 150 candidate grasps at various spatial positions for each target object.

## 4.3 GRASP GUIDANCE SELECTION

We generate diverse grasp candidates using the method in Section 4.2. However, not all candidates are feasible for fast grasping execution. Grasps on the object's rear side may be kinematically constrained or occluded, preventing real-time completion. Therefore, we design a selection process to identify valid and stable grasp proposals.

Our method employs hierarchical filtering from two perspectives: **Executability filtering**: We filter grasps requiring the arm to circumvent the object or follow extended trajectories. Specifically, based on the relative pose between the robot and the target object, we define a forward grasping space for grasp planning: namely, a grasping cone benchmarked against the normal vector pointing from the target towards the robot. Grasping poses within this space, referred to as forward grasps, are characterized by the shortest motion paths and minimal joint movement, thereby achieving the highest grasping efficiency. In contrast, non-forward grasps (such as rear grasps) require the end-effector to undergo rotations exceeding 90 degrees and perform avoidance maneuvers, which significantly increases motion complexity and time cost. Consequently, the grasps that require end-effector rotations exceeding 90 degrees are systematically filtered during the planning phase, as these significantly increase execution time and violate fast grasping requirements. **Collision detection filtering**: We eliminate candidates where the end-effector would descend below the object's lowest point to prevent collisions with supporting surfaces.

Following the initial filtering, we propose a computational approach based on hand envelopment degree to evaluate grasp stability and dynamic execution capability. The core advantage of this method lies in requiring no explicit surface normal information, effectively handling partial or noisy point cloud data, and significantly improving computational efficiency for real-time applications.

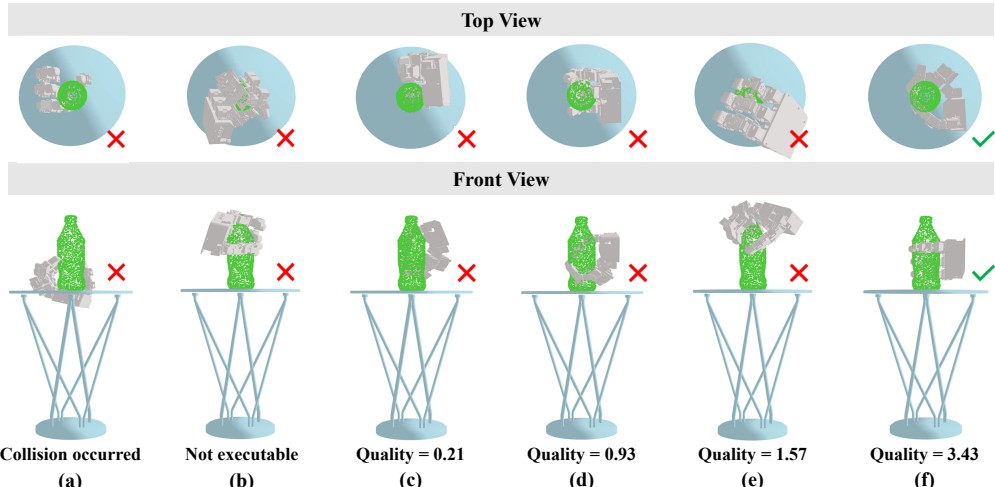

Figure 4: Grasp Guidance Selection

We quantify grasp quality through two complementary metrics: Grasp Width Coverage (GWC) and Grasp Depth Coverage (GDC).

**Grasp Width Coverage (GWC)** quantifies the coverage degree of the grasping interval vector formed between the fingers and thumb relative to the object's maximum effective width along this vector. This metric is grounded in the mechanical principle that grasp stability increases when contact points are positioned on opposing sides of the object, thereby maximizing force closure and resistance to external torques, as shown in Fig.4 . We identify the positions of all finger tips represented by their respective 3D coordinates. We define the finger grasp axis as the vector $[\mathbf{f}_k, \mathbf{f}_{thumb}]$ connecting the thumb $\mathbf{f}_{thumb}$ and finger $\mathbf{f}_k$, which indicates the coverage range of the grasping gesture between the thumb and that specific finger. Formally, let $\mathbf{w}_k$ denote the corresponding unit direction, and let $f_k^w = \mathbf{w}_k^\top \mathbf{f}_k$ and $f_{thumb}^w = \mathbf{w}_k^\top \mathbf{f}_{thumb}$ be the scalar projections of the two fingertips onto this axis. The object's maximum effective width is defined as the distance between the two farthest points of the object point cloud when projected onto the direction of $[\mathbf{f}_k, \mathbf{f}_{thumb}]$, which is identified as $[p_{min}, p_{max}]$ in this 1D projected space. Then the GWC for finger $k$ is:

$$\text{GWC}_k = \frac{\left|\left[f_k^w, f_{thumb}^w\right] \cap [p_{min}, p_{max}]\right|}{\left|[p_{min}, p_{max}]\right|} \tag{2}$$

where "$|[a,b]|$" denotes the length of a 1D interval and intersection "$\cap$" is taken in this scalar space.

**Grasp Depth Coverage (GDC)** quantifies the coverage degree of the grasping interval formed between the fingers and the palm reference plane relative to the object's maximum effective depth. A greater depth value indicates that the object is positioned deeper within the hand's natural grasping enclosure. This deeper placement significantly increases the likelihood of contact not only with the fingertips but also with the broader palmar regions (such as the finger pads and the central palm). Consequently, it facilitates multi-point, distributed contact, which enhances grasp robustness, as shown in Fig.4 . We define the vector $[\mathbf{d}_k, \mathbf{d}_{palm}]$ as the projection of the vector from the palm center to the finger position onto the direction perpendicular to the palm plane, representing the finger reachable depth. Let $\mathbf{n}_{palm}$ denote the unit normal vector of the palm plane. We compute the scalar depths $d_k = \mathbf{n}_{palm}^\top \mathbf{d}_k$ and $d_{palm} = \mathbf{n}_{palm}^\top \mathbf{d}_{palm}$ along this direction, and in Eq. (3) the notation $[\mathbf{d}_k, \mathbf{d}_{palm}]$ refers to the 1D depth interval $[d_{palm}, d_k]$ in this depth space. This indicates the maximum theoretical grasping depth that the finger structure can provide in a specific grasp configuration. We identify $[d_{min}, d_{max}]$ as the interval spanned by the distance between the two farthest points of the object point cloud when projected onto the direction of $[\mathbf{d}_k, \mathbf{d}_{palm}]$. Then the GDC for finger $k$ is:

$$\text{GDC}_k = \frac{\left|\left[d_{palm}, d_k\right] \cap [d_{min}, d_{max}]\right|}{\left|[d_{min}, d_{max}]\right|} \tag{3}$$

The overall grasp quality combines coverage information from both width and depth dimensions:

$$\text{Quality} = \left( \sum_{k=1}^{3} \text{GWC}_k \cdot \text{GDC}_k \right) \cdot \text{GDC}_{thumb} \tag{4}$$

where the thumb's grasping depth serves as a critical factor influencing overall grasp stability. We select the grasp with the highest Quality value as the guidance.As demonstrated in Fig.4, the quality metric shows strong visual-quantitative correspondence.

## 4.4 POLICY LEARNING

We employ reinforcement learning to train a policy that controls the robot's full-body motion, leveraging real-time in-hand tactile information and grasp guidance derived from the aforementioned methodology.

**Observation**. The observation space of the policy is represented as $\mathbf{O}_t = (\mathbf{q}_t, \mathbf{v}_t, \mathbf{o}_t, \mathbf{c}_t, \mathbf{g})$, where $t$ denotes the current time step, $\mathbf{q}_t = \{\text{qpos}_{arm,t}, \text{qpos}_{hand,t}\}$ represents the robot state including arm joint configuration $\text{qpos}_{arm,t}$, and hand joint configuration $\text{qpos}_{hand,t}$, $\mathbf{v}_t = \{\mathbf{v}_{f,t}, \mathbf{v}_{y,t}\}$ comprises the forward velocity $\mathbf{v}_{f,t}$ and yaw velocity $\mathbf{v}_{y,t}$ of the mobile base, $\mathbf{o}_t = \{\text{pos}_{obj,t}, \text{dis}_{obj,t}\}$ contains the object coordinates $\text{pos}_{obj,t}$ in the wrist camera coordinate system and the distance $\text{dis}_{obj,t}$ between the object and the palm center, $\mathbf{c}_t$ represents the binary contact states of the hand, with contact detection implemented via pressure-sensitive tactile sensors, and $\mathbf{g} = \{\text{pos}_g, \text{rot}_g, \text{qpos}_g\}$ constitutes the grasp guidance, which includes the position vector $\text{pos}_g \in \mathbb{R}^3$, the hand rotation $\text{rot}_g \in \mathbb{R}^3$ (represented using Euler angles), and the joint configuration $\text{qpos}_g \in \mathbb{R}^{13}$.

**Action**. The policy outputs actions $\mathbf{A}_t = \{\mathbf{v}_{base,t}, \text{qpos}_{arm,t}, \text{qpos}_{hand,t}\}$, where $\mathbf{v}_{base,t} \in \mathbb{R}^2$ contains the forward and yaw velocities for mobile base, $\text{qpos}_{arm,t} \in \mathbb{R}^5$ represents the 5-DOF arm joint positions(the fourth joint fixed for stability), and $\text{qpos}_{hand,t} \in \mathbb{R}^{13}$ specifies 13-DOF hand joint positions.

**Reward Design**. We design a multi-stage reward function to guide the policy through the complete grasping pipeline:

$$R_{sum} = \lambda_L \cdot R_{locomotion} + \lambda_P \cdot R_{preparation} + \lambda_E \cdot R_{execution} \tag{5}$$

where $R_{locomotion}$ comprises arm operation radius reward $R_{radius}$, base movement velocity reward $R_{move}$, base orientation reward $R_{ori}$, and height reward $R_{height}$ to encourage efficient whole-body coordination for approaching objects; $R_{preparation}$ includes pre-grasp reward $R_{pre-grasp}$ and hand joint configuration alignment reward $R_{hand}$ to align the end-effector and hand with the grasp guidance; and $R_{execution}$ consists of rapid completion reward $R_{reach}$, stable grasping reward $R_{stable}$, and tactile reward $R_{tactile}$ to leverage tactile feedback and ensure stable grasping during high-speed motion.

Notably, while the generated grasps are geometrically enveloping (as shown in Sec. 4.3), they do not guarantee successful object retention. Therefore, we introduce a tactile reward to encourage active hand-object contact during grasping:

$$R_{\text{tactile}} = \begin{cases} f_{\text{touch}}, & \text{if } c_{\text{palm}} \neq 0, \\ 0, & \text{otherwise,} \end{cases} \tag{6}$$

where $c_{\text{palm}} \in \{0, 1\}$ denotes the binary contact flag of the robotic palm detected through pressure sensors, and $f_{\text{touch}} \in \mathbb{N}$ represents the total number of finger contacts detected via pressure-based tactile sensors, calculated as $f_{\text{touch}} = \sum_{i=1}^{N} c_i$ across all finger segments. Other detailed reward formulations are provided in Appendix A.

**Policy Optimization**. We train our grasping policy using Proximal Policy Optimization (PPO) Schulman et al. (2017), which maximizes the following clipped surrogate objective:

$$L^{\text{CLIP}}(\theta) = \mathbb{E}_t \left[ \min \left( r_t(\theta)\hat{A}_t, \text{clip}(r_t(\theta), 1 - \varepsilon, 1 + \varepsilon)\hat{A}_t \right) \right] \tag{7}$$

where $r_t(\theta) = \frac{\pi_\theta(a_t|s_t)}{\pi_{\theta_{\text{old}}}(a_t|s_t)}$ denotes the probability ratio between current and old policies, and $\hat{A}_t$ represents the advantage estimate computed via Generalized Advantage Estimation (GAE).Detailed hyperparameter settings and network architectures are provided in Appendix B.

# 5 EXPERIMENTS

## 5.1 EXPERIMENT SETTINGS

**Dataset.** We utilize two types of datasets: First, to foster diversity in the generator's learned grasping strategies, we create a grasp synthesis dataset for training the grasp generation model, comprising 478,200 validated grasp poses spanning 4,782 object instances with 100 distinct grasps per object, synthesized using a methodology similar to (Chen et al., 2025). These synthesized grasps feature tight finger and palm configurations with full hand-object contact, as visualized in Appendix C. Second, for policy training and evaluation, we use Realdex objects (Liu et al., 2024) with 53 object instances split into training instances (34) and testing instances (19), where testing instances are further categorized into easy-to-grasp cases (11) and hard-to-grasp cases (8) based on geometric complexity. For real-world experiments, we prepare 19 distinct objects with variations in size, shape, and weight, divided into 11 simple objects and 8 complex objects as shown in Appendix E.

**Training and Experimental Setup.** We first train the generator exclusively on the synthetic dataset, then freeze its parameters for inference to generate diverse grasps from which the optimal one is selected as policy input and contributes to reward computation. We train the policy across 48 parallel environments, with training conducted on an Intel i9 14900K CPU and NVIDIA GeForce RTX 4090 GPU. In simulation, the basic timestep frequency is set to 200 Hz, with policy inference executed every 10 simulation steps, corresponding to a 20 Hz control frequency. When deploying the policy on the real robot, we maintain the same 20 Hz control frequency to ensure sim-to-real consistency. The computational time breakdown for each system component is presented in Appendix F.

**Evaluation Metric.** (i) Success Rate:Success is defined as lifting the object without slippage or drops, with the object remaining held in hand until the episode's maximum time, which is set to 2 seconds. (ii) Hand-object offset distance: The distance the object slides on the hand when the hand makes contact with it.

Table 1: **Success rate (%) and Hand-object offset distance(cm) of unseen objects in simulation.** Each object undergoes $10\times1000$ grasp trials. TS: two-stage; FS: contact-force direction selection; ES: hand-envelopment degree selection.Values outside and inside the parentheses denote the success rate and the hand-object offset distance, respectively.

| Method | Full Point Cloud | | | Partial Point Cloud | | |
|---|---|---|---|---|---|---|
| | Easy (11) | Hard (8) | Average | Easy (11) | Hard (8) | Average |
| Burgess-Limerick et al. (2023) | 15.11(7.56) | 8.33(8.13) | 12.25(7.8) | 15.11(7.56) | 8.33(8.13) | 12.25(7.8) |
| One-Stage (OS) | 9.13(7.42) | 8.87(7.69) | 9.02(7.53) | 0(-) | 0(-) | 0(-) |
| TS + FS | 86.43(1.75) | 30.60(3.01) | 62.92(2.28) | 3.59(7.99) | 4.88(7.91) | 4.13(7.95) |
| Ours (TS + ES) | **89.21(1.28)** | **32.73(2.71)** | **65.42(1.88)** | **62.17(2.67)** | **25.09(3.97)** | **46.55(3.21)** |

## 5.2 RESULTS ON FAST GRASPING

We compare our method with baselines in simulation using both full and partial object point clouds. We compare against three baselines: (i) The reactive mobile manipulation architecture from (Burgess-Limerick et al., 2023), which provides an optimized framework for on-the-move grasping; (ii) One-stage method that takes point cloud features as input, where we utilize point features extracted by PointNet (Qi et al., 2017) as input without guidance; (iii) Two-stage method utilizing force-direction-based selection (Ciocarlie & Allen, 2009), where we use grasp candidates selected by the force-direction-based quality metric as guidance.

Table 2: **Ablation Study**. Success Rate (S.R.) and Hand-object offset distance(cm) for objects with varying levels of difficulty, with each object subjected to $10\times500$ grasp trials.Values outside and inside the parentheses denote the success rate and the hand-object offset distance, respectively.

| w/o | Reward | | | Tactile | | Grasp Guidance Selection | | | Ours |
|---|---|---|---|---|---|---|---|---|---|
| | $R_{radius}$ | $R_{height}$ | $R_{pre\text{-}grasp}$ | $R_{tactile}$ | Tactile_obs | GDC+GWC | GDC | GWC | |
| Easy case(11) | 84.44(0.91) | 12.14(9.56) | 46.93(5.38) | 28.36(3.19) | 82.84(1.41) | 0(-) | 5.31(7.17) | 81.81(1.52) | **89.21(1.28)** |
| Hard case(8) | 34.35(1.79) | 4.98(7.59) | 15.96(8.42) | 15.86(4.92) | 27.65(3.04) | 0(-) | 8.70(7.09) | 29.76(3.16) | **32.73(2.71)** |
| Average | 63.34(1.28) | 9.12(8.73) | 33.76(6.66) | 23.09(3.91) | 59.60(2.09) | 0(-) | 6.73(7.13) | 59.89(2.21) | **65.42(1.88)** |

Table 3: **Real-world success rate.** Each object tested in 10 consecutive trials.

| Velocity Constraints | $\max(\mathbf{v}_f) = 1.3m/s$, $\max(\mathbf{v}_y) = 1.0rad/s$ | | | $\max(\mathbf{v}_f) = 0.65m/s$, $\max(\mathbf{v}_y) = 0.5rad/s$ | | |
|---|---|---|---|---|---|---|
| S.R. (%) | **Easy case(11)** | **Hard case(8)** | **Average** | **Easy case(11)** | **Hard case(8)** | **Average** |
| ours (W/O DR) | 0 | 0 | 0 | 0 | 0 | 0 |
| ours (W/O LPF) | 0 | 0 | 0 | 0 | 0 | 0 |
| ours | 27 | 12 | 20.68 | 32 | 16 | 25.26 |

As shown in Table 1, our method demonstrates significant improvements by consistently achieving the highest success rates across both full and partial point cloud conditions, highlighting its robustness and superiority. Under the full point cloud setting, our method significantly outperforms all baselines. Moreover, compared to the two-stage force-direction-based selection method (TS+FS), our envelopment-based selection (TS+ES) shows a clear advantage. More importantly, our approach exhibits remarkable robustness to partial observability, while all baseline methods suffer severe performance degradation. It is worth noting that the method of (Burgess-Limerick et al., 2023) treats objects as a single coordinate point without point cloud input, hence its identical results under both point cloud conditions and its inferior performance compared to our method. These results underscore the critical advantage of our envelopment-based selection strategy in generating high-quality grasp guidance, particularly in realistic scenarios with incomplete perceptual data.

### 5.3 ABLATION STUDY

We first evaluate the effectiveness of our reward design (Section 4.4) by comparing against the following variants, as shown in Table 2: (i) without the arm operation radius reward (w/o $R_{\text{radius}}$) to verify the effectiveness of encouraging larger arm extension, (ii) without the height reward (w/o $R_{\text{height}}$) to verify the effectiveness of avoiding collisions with the table beneath the object, (iii) without utilizing the guidance translation and rotation (w/o $R_{\text{pre-grasp}}$) to verify the effectiveness of providing direct joint angle guidance, (iv) without the tactile reward (w/o $R_{\text{tactile}}$) to validate the effectiveness of encouraging hand-object contact, and (v) without binary tactile input in the observation space (w/o Tactile_obs) to quantify its contribution. Results show that removing tactile observations reduces success by 9%, while omitting the tactile reward leads to a 65% drop, confirming the critical role of tactile feedback in slip prevention.

We then evaluate the effectiveness of our grasp guidance selection method as described in Section 4.3. As shown in Table 2, we present the results obtained without employing any filtering strategy (w/o GWC+GDC), where a grasp candidate is randomly selected from the generated grasp proposals to serve as guidance, and the results of the method without GWC and without GDC, respectively.

### 5.4 REAL ROBOT DEPLOYMENT

**Sim2Real Transfer.** Due to the inherent complexity of our system, a significant sim-to-real gap remains. To mitigate this discrepancy as effectively as possible, we incorporate the following techniques: (i) **Object observation**: In simulation, privileged information such as object point clouds and poses can be directly accessed. In the real world, however, object data must be extracted from sensory observations through segmentation. To bridge this gap, we attached color-based markers to physical objects, enabling efficient color-driven segmentation. This approach preserves geometric completeness while significantly reducing perception latency, thereby improving the inference efficiency of the model. (ii) **Low-Pass Filter**: In simulation, robots can execute high-frequency control commands instantaneously. However, in the real world, due to physical inertia, limited motor response speed, and performance constraints of low-level controllers (e.g., PID), executing such highly oscillatory commands becomes infeasible. To bridge this gap, we apply a Low-Pass Filter (LPF) to smooth the control commands for the mobile base, robotic arm, and hand. The filtering process can be summarized by the following first-order recursive formula:

$$c_{\text{filtered}}[t] = \alpha \cdot c_{\text{filtered}}[t-1] + (1-\alpha) \cdot c_{\text{raw}}[t] \tag{8}$$

where $c$ is the control command indicating the velocity command or joint position command, $\alpha = 0.3$ is the smoothing coefficient, $c_{\text{raw}}[t]$ denotes the raw command at time step $t$, and $c_{\text{filtered}}[t]$ represents the filtered output. (iii) **Domain Randomization**(DR): Domain randomization has been proven to be effective in sim2real transfer (Tobin et al., 2017). To enhance the robustness of the

model in the real world, we not only randomize environmental settings (Section 3) but also incorporate observation noise and action noise, thereby improving the policy's adaptability to sensor errors and actuator uncertainties in real-world conditions.

**Real-world Results.** Table 3 demonstrates that domain randomization (DR) and low-pass filtering (LPF) are both essential for real-world deployment, with either component's absence causing complete failure. Our system achieves 20.68% success under high-speed conditions and 25.26% under reduced-speed conditions, showing that velocity reduction enhances stability. Given the complexity of coordinating 20+ DOF for high-speed dynamic grasping amid impact forces, these results highlight the challenging nature of this task. Success occurs only with the complete configuration, confirming our sim-to-real transfer approach.

## 6 CONCLUSION

This paper presents a learning-based framework for mobile fast grasping, which addresses stability challenges in high-speed operations by integrating whole-body control, grasp guidance, and tactile feedback. The proposed two-stage policy first generates diverse grasp candidates and then executes coordinated motions guided by optimal selection. A unified reinforcement learning framework enables simultaneous control of the mobile base, robotic arm, and dexterous hand, while tactile feedback facilitates real-time grasp adjustment and improves generalization across diverse object geometries. Extensive experiments in both simulation and real-world settings demonstrate the superior performance of our approach, along with effective ~~zero-shot~~ sim-to-real transfer.

## 7 LIMITATIONS AND FUTURE WORK

There are several limitations that suggest directions for future work. First, obstacle avoidance is currently handled only in relatively simple scenarios; extending the approach to cluttered and highly complex environments remains challenging. Second, the system still struggles with flat or otherwise specially shaped objects, and safety during high-speed motion in complex environments is not yet fully guaranteed. Future work will focus on optimizing grasping strategies for such challenging object geometries, enhancing safety mechanisms for dynamic, high-speed operation, and further exploring control paradigms that more tightly integrate fast navigation with robust manipulation. These improvements are expected to enhance the practicality and adaptability of the system in real-world environments.

### ETHICS STATEMENT

We are committed to maintaining transparency and integrity throughout the research process. For safety and ethical considerations, our experimental procedure was strictly divided into simulation and real-world phases. The training and initial validation of the method were conducted entirely within a simulated environment. The approach was transitioned to physical hardware testing only after its performance was deemed reliable in simulation. During all real-world experiments, a human safety operator was present to continuously monitor the robot's motions. This operator was authorized to proactively halt the operation via an emergency stop mechanism at the first sign of any potential hazard.

### REPRODUCIBILITY STATEMENT

To ensure the reproducibility of our work, we commit to open-sourcing the complete codebase for training and inference, as well as the pre-trained weights for our best-performing model, this release will occur upon the paper's acceptance. The training dataset will be made publicly available upon paper acceptance, along with detailed data collection methodologies to ensure reproducibility. Besides, the implementation details are presented in the main text, while the comprehensive configuration parameters are provided in the supplementary material.

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

# Appendix

This supplementary material offers comprehensive technical details and extended experimental results for the proposed fast mobile grasping framework. We organize the content as follows:

- Section A details the multi-stage reward design and reward curves.

- Section B lists hyper-parameters and implementation specifics.

- Section C visualizes the synthetic grasp dataset.

- Section D illustrates the grasp-guidance selection process.

- Section E displays the 19 real-world test objects grouped by geometric complexity.

- Section F gives the runtime breakdown on an RTX 4090.

- We conclude with a statement on the limited, language-polishing use of large-language models during manuscript preparation.

## A    DETAILED REWARD DESIGN

We design a comprehensive reward function to encourage fast and safe grasping behavior. The reward is formulated based on the object position $\mathbf{pos}_{obj}$, hand joint configuration $\mathbf{qpos}_{hand}$, end-effector position $\mathbf{pos}_{ee}$, arm operation radius $r_{arm}$, rotation $\mathbf{rot}_{ee}$ and velocity $\mathbf{v}_{ee}$, guidance position $\mathbf{pos}_g$, guidance rotation $\mathbf{rot}_g$, and guidance joint configuration $\mathbf{qpos}_g$. Here, $r_{arm}$ represents the arm operation radius, which is determined by the distance between the end-effector and the arm base position. The reward function is decomposed into three stages to systematically guide the grasping process:

$$R_{sum} = \lambda_L \cdot R_{locomotion} + \lambda_P \cdot R_{preparation} + \lambda_E \cdot R_{execution} \tag{9}$$

where $\lambda_L$, $\lambda_P$, and $\lambda_E$ are weighting coefficients for each stage.

### A.1    LOCOMOTION STAGE REWARD ($R_{locomotion}$)

This component encourages efficient whole-body coordination for approaching objects beyond the robot's initial reach:

- **Arm operation radius reward**: To encourage larger arm extension and increase base mobility for responding to sudden motion changes: $R_{radius} = \|r_{arm}\|_2$.

- **Base movement velocity reward**: $R_{move} = \|\mathbf{v}_f\| + \|\mathbf{v}_y\|$, where $\mathbf{v}_f$ is the forward velocity and $\mathbf{v}_y$ is the yaw velocity.

- **Base movement orientation reward**: To encourage the robotic arm to manipulate the object at its maximum operable radius: $R_{ori} = -\|\omega\|$, where $\omega$ is the angle between the target direction and its velocity direction, the target direction is set as the direction from the robot to a point that is offset from $\mathbf{pos}_g$ by the $r_{arm}$ along the robot's initial horizontal heading.

- **Height reward**: To encourage the hand to maintain a certain height to avoid collisions with the table beneath the object:

$$R_{\text{height}} = \begin{cases} -\|\mathbf{pos}_{ee-h} - \mathbf{pos}_{obj-h}\|_2, & \|\mathbf{pos}_g - \mathbf{pos}_{ee}\|_2 > 0.5, \\ 0, & \text{otherwise}, \end{cases} \tag{10}$$

where $\mathbf{pos}_{obj-h}$ is the height of the object and $\mathbf{pos}_{ee-h}$ is the height of the end-effector.

## A.2   PREPARATION STAGE REWARD ($R_{preparation}$)

This component aligns the end-effector and hand with the grasp guidance to prepare for optimal grasping:

- **Pre-grasp reward**: Inspired by Dasari et al. (2022), we design a pre-grasp reward to encourage the policy to align the robot's end-effector pose with the guided position and orientation. To provide more direct and actionable optimization signals, the guided pose is transformed into the corresponding target joint configuration. At the onset of grasping, $R_{\text{pre-grasp}}$ is defined as:

$$R_{\text{pre-grasp}} = \begin{cases} -\alpha_{rot} \left\| \mathbf{rot}_{ee} - \mathbf{rot}_g \right\| - \alpha_{pos} \left\| \mathbf{pos}_{ee} - \mathbf{pos}_g \right\|_2, & \left\| \mathbf{pos}_g - \mathbf{pos}_{ee} \right\|_2 \leq 0.1, \\ 0, & \text{otherwise}, \end{cases}$$ (11)

  where $\alpha_{\text{rot}}, \alpha_{\text{pos}} > 0$ are weighting factors.

- **Hand joint configuration alignment reward**: To encourage hand opening when the current state exceeds the threshold and promote hand closing when it is within the threshold. When grasping will occur, the $R_{\text{hand}}$ is defined as:

$$R_{\text{hand}} = \begin{cases} \frac{1}{1+2\cdot\left\|\mathbf{qpos}_{hand}\right\|_2}, & \left\| \mathbf{pos}_g - \mathbf{pos}_{ee} \right\|_2 > 0.2, \\ \frac{5}{1+2\cdot\left\|\mathbf{qpose}_{hand} - \mathbf{qpose}_g\right\|_2}, & \text{otherwise}, \end{cases}$$ (12)

## A.3   EXECUTION STAGE REWARD ($R_{execution}$)

This component leverages tactile feedback to ensure stable grasping during high-speed motion:

- **Rapid completion reward**: To encourage fast execution velocity when approaching the target and swift completion of the grasping task followed by prompt withdrawal from the table surface:

$$R_{\text{reach}} = \begin{cases} e^{\mathbf{v}_{ee}}, & \left\| \mathbf{pos}_g - \mathbf{pos}_{ee} \right\|_2 \leq 0.25, \\ 0, & \text{otherwise}, \end{cases}$$ (13)

- **Stable grasping reward**: This reward is computed based on the duration for which the object is held by the hand: $R_{\text{stable}} = 2 \cdot \Delta t_{\text{grasp}}$.

- **Tactile reward**: To encourage active hand-object contact during grasping:

$$R_{\text{tactile}} = \begin{cases} f_{\text{touch}}, & \text{if } c_{\text{palm}} \neq 0, \\ 0, & \text{otherwise}, \end{cases}$$ (14)

  where $c_{\text{palm}} \in \{0, 1\}$ denotes the binary contact flag of the robotic palm detected through pressure sensors, and $f_{\text{touch}} \in \mathbb{N}$ denotes the total number of finger contacts detected via pressure-based tactile sensors, calculated as the sum of binary contact flags across all finger segments: $f_{\text{touch}} = \sum_{i=1}^{N} c_i$.

Fig.A illustrates the training reward curves over episodes, demonstrating the convergence behavior of our multi-stage reward design. The red curve shows the total reward progression, while the blue curves represent individual reward components across the three training stages. The locomotion stage rewards (arm operation radius, base movement velocity, base movement orientation, and height reward) show steady improvement as the policy learns efficient whole-body coordination. The preparation stage rewards (pre-grasp and hand joint configuration alignment) exhibit convergence patterns that reflect the policy's ability to align with grasp guidance. The execution stage rewards (rapid completion, stable grasping, and tactile reward) demonstrate increasing values as the policy masters contact-rich manipulation skills. These curves collectively validate the effectiveness of our hierarchical reward structure in guiding the learning process from initial approach to successful grasp completion.

## B  HYPERPARAMETER CONFIGURATION AND IMPLEMENTATION DETAILS

**Grasp Generator Architecture.** Our grasp generator employs a Conditional Variational Autoencoder (CVAE) architecture. A PointNet network with layer dimensions [64, 128, 128] extracts global feature vectors from input point clouds, serving as conditioning information. The CVAE encoder consists of an MLP with dimensions [64, 128, 256], while the decoder processes concatenated latent vectors and conditions through an MLP with sizes [384, 256, 256, 128]. All layers utilize ReLU activation functions. Training is conducted with batch size 400 over 2000 epochs using the Adam optimizer with learning rate $1 \times 10^{-3}$, KL divergence weight 0.005, qpos weight 8, and translation weight 10.

**Policy Network Configuration.** The policy network is implemented as an MLP with layer sizes [1024, 512, 512, 256, 128, 128, 128, 64, 64], employing ELU activation functions. The value network shares identical architecture for state value estimation in advantage computation. Training utilizes 48 parallel simulation environments with PPO hyperparameters: learning rate $1 \times 10^{-3}$, discount factor 0.99, GAE parameter 0.95, clipping parameter 0.2, and entropy coefficient $1 \times 10^{-2}$. The policy undergoes 72,000 iterations with mini-batch size 4 and 5 epochs per iteration.

**Reward Function Weights.** The multi-stage reward function employs coefficients $\lambda_L$, $\lambda_P$, and $\lambda_E$ for locomotion, preparation, and execution stages respectively. Specifically: $\lambda_L = [\lambda_{radius}, \lambda_{move}, \lambda_{ori}, \lambda_{height}]$ weights the radius, movement, orientation, and height rewards; $\lambda_P = [\lambda_{pre-grasp}, \lambda_{hand}]$ weights the pre-grasp and hand configuration rewards; $\lambda_E = [\lambda_{reach}, \lambda_{stable}, \lambda_{tactile}]$ weights the reaching, stability, and tactile rewards. Individual coefficient values are detailed in Table 4.

Table 4: **Coefficients of each reward function.**

| Weight | Value |
|---|---|
| $\lambda_{radius}$ (for arm operation radius reward) | 0.0001 |
| $\lambda_{move}$ (for base movement velocity reward) | 2 |
| $\lambda_{ori}$ (for base movement orientation reward) | 20 |
| $\lambda_{height}$ (for height reward) | 4 |
| $\lambda_{pre-grasp}$ (for pre-grasp reward) | 5 |
| $\lambda_{hand}$ (for hand joint configuration alignment reward) | 2 |
| $\lambda_{reach}$ (for rapid completion reward) | 5 |
| $\lambda_{stable}$ (for stable grasping reward) | 5 |
| $\lambda_{tactile}$ (for tactile reward) | 1 |
| $\alpha_{rot}$ (for rotation weighting in pre-grasp reward) | 2 |
| $\alpha_{pos}$ (for position weighting in pre-grasp reward) | 0.001 |

## C  GRASP PROPOSAL GENERATION VISUALIZATION

Fig. 6 illustrates a subset of the 4,782 object instances and their corresponding diverse grasp poses generated using our synthesis pipeline. The visualization exemplifies the key characteristics of the dataset, including the coverage of various object geometries and the quality of the synthesized grasps featuring tight configurations and full contact, which are crucial for training a diverse and effective grasp proposal generator.

## D  GRASP GUIDANCE SELECTION VISUALIZATION

Fig. 7 provides further visualizations of grasp guidance selection. Multiple grasp poses are compared from both top and front views, with annotations indicating collision occurrence, executability, and the corresponding grasp quality scores. It can be observed that as the grasp quality score increases, the poses gradually become more stable, collision-free, and executable, which fully demonstrates the effectiveness and rationality of our method in guiding optimal grasp selection.

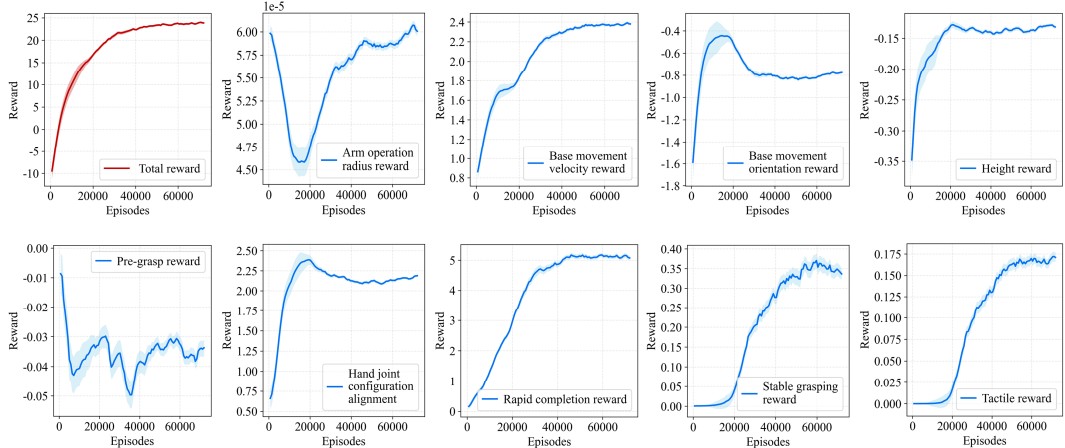

Figure 5: Training reward curves over episodes. The red curve shows the total reward, while blue curves represent individual reward components.

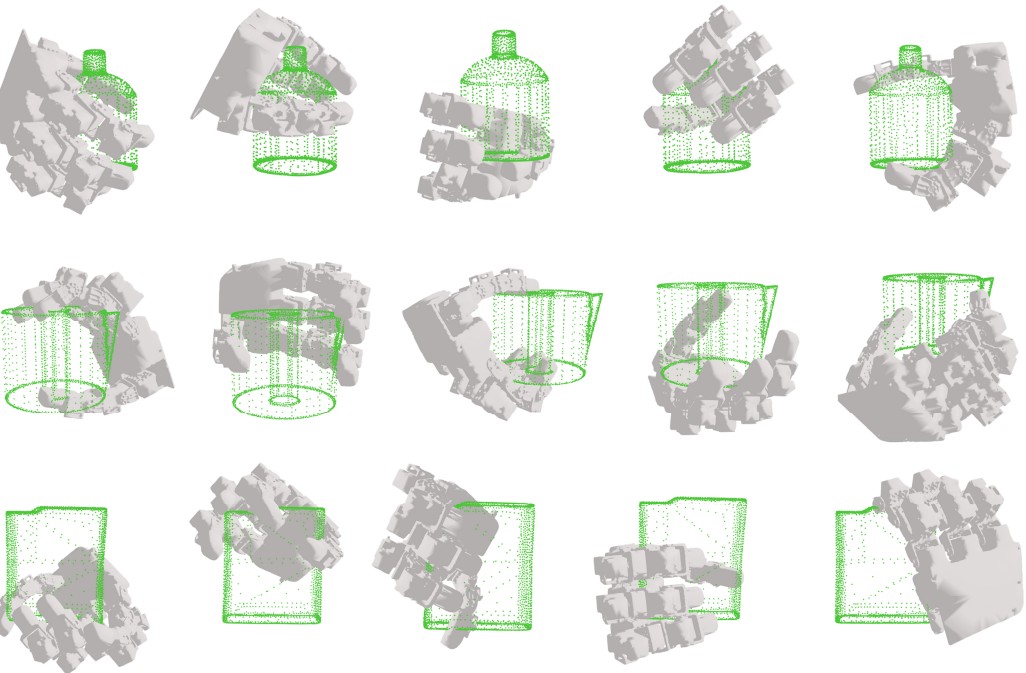

Figure 6: Visualization of the synthetic grasp dataset used for training the grasp proposal generator

## E   REAL-WORLD TEST OBJECTS VISUALIZATION

Fig. 8 shows the 19 objects used in the physical experiments. The left 11 items form the Simple set (boxes, cylinders, and other regular shapes), while the right 8 items constitute the Complex set (flat, thin, or irregular objects). Together they span variations in size, shape, and weight, allowing us to evaluate grasping performance under diverse geometric complexities in the real world.

## F   COMPUTATIONAL TIME BREAKDOWN

Table 5 reports the per-component latency of a single inference pass, measured on an NVIDIA RTX 4090 GPU . Grasp-proposal generation takes 0.46 ms and policy inference 0.33 ms; the quality-

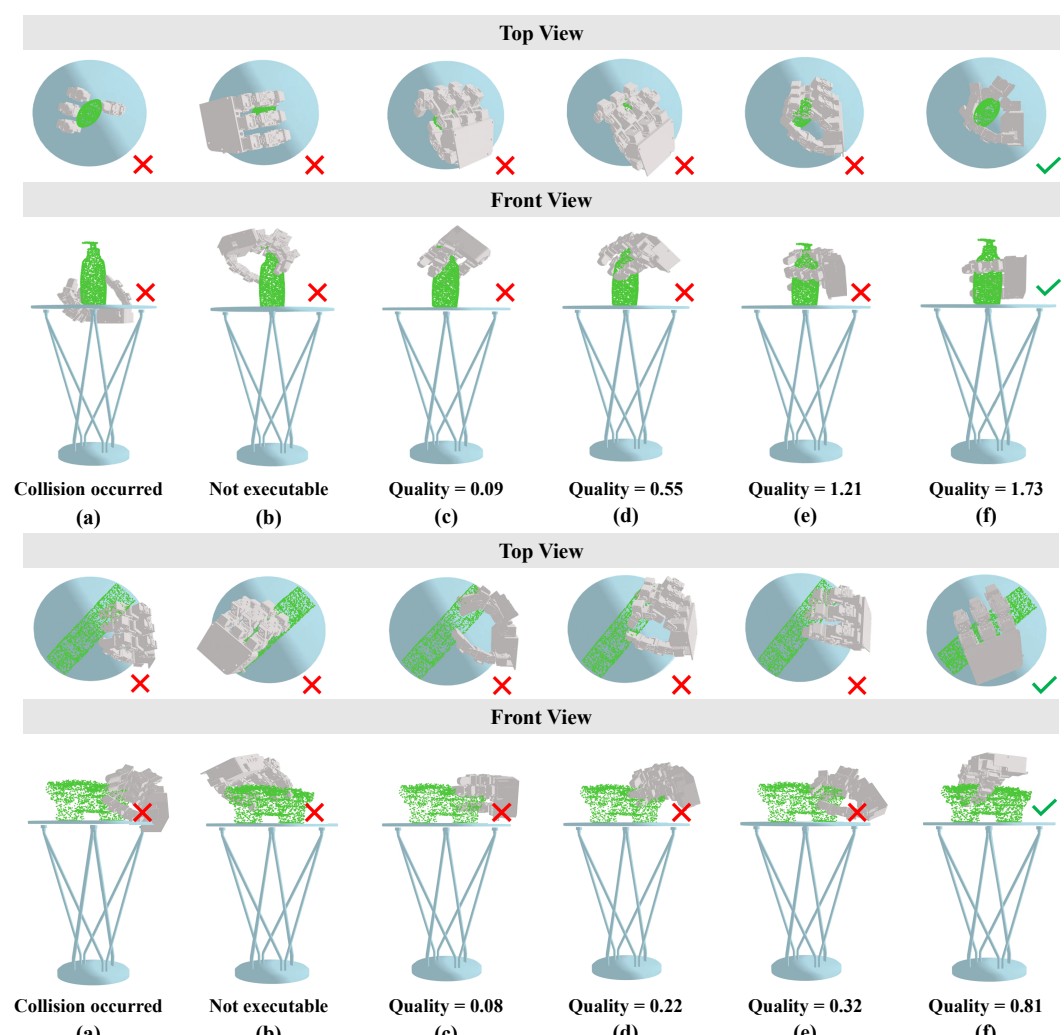

Figure 7: Grasp Guidance Selection Visualization

Table 5: **Computational time breakdown for each system component.** All measurements are conducted on an NVIDIA GeForce RTX 4090 GPU with single-batch inference.

| System Component | Computation Time (ms) |
| --- | --- |
| Grasp Proposal Generation | 0.460 |
| Grasp Guidance Selection | 2.877 |
| Policy Prediction | 0.332 |
| **Total** | 3.669 |

guided selection of the best pose dominates at 2.88 ms. The entire pipeline completes in 3.67 ms, meeting the real-time requirement of the system.

## G   THE USE OF LARGE LANGUAGE MODELS

In the preparation of this manuscript, Large Language Models (LLMs) were utilized solely to aid in the writing and polishing of the manuscript. Specifically, AI-assisted tools were employed to enhance the clarity, conciseness, and overall fluency of the academic writing. This involved sugges-

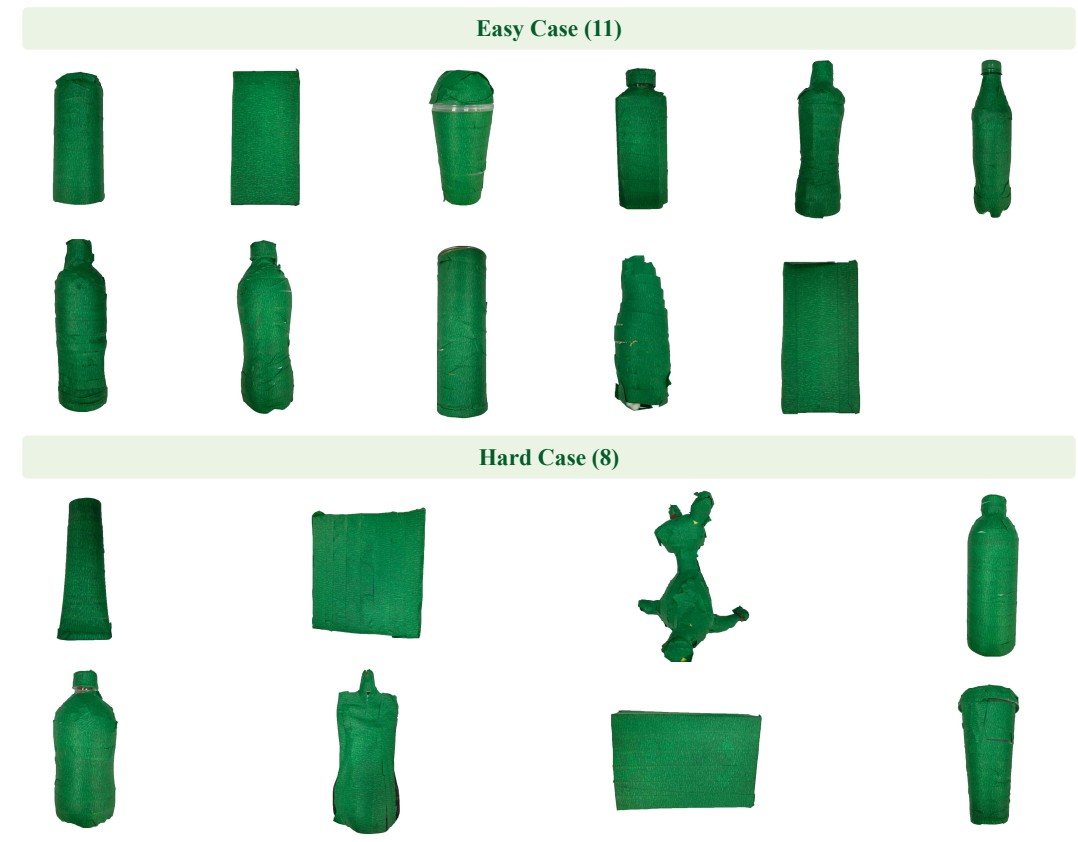

Figure 8: Real-world test objects

tions for grammatical corrections, stylistic improvements in phrasing, and ensuring consistency in technical terminology.

It is important to note that the LLM was not involved in the ideation, research methodology, or experimental design. All research concepts, ideas, and analyses were developed and conducted by the authors. The contributions of the LLM were solely focused on improving the linguistic quality of the paper, with no involvement in the scientific content or data analysis. The authors take full responsibility for the content of the manuscript, including any text generated or polished by the LLM. We have ensured that the LLM-generated text adheres to ethical guidelines and does not contribute to plagiarism or scientific misconduct.

