# OpenReview forum: "FastGrasp: Learning-based Whole-body Control method for Fast Dexterous Grasping with Mobile Manipulators"
_ICLR.cc/2026/Conference — Submitted to ICLR 2026_

### Official Review · Reviewer_RjXK · 2025-10-26

**Soundness:** 2
**Presentation:** 3
**Contribution:** 2
**Rating:** 4
**Confidence:** 4

**Summary:**

This paper proposes a two-stage RL-based framework for object grasping with mobile manipulators. A cVAE model generates grasping poses based on object observations, and an RL policy controls the robot to reach and grasp the moving object. The sim-to-real gap for privileged observation is handled by attaching markers to the object for segmentation.

**Strengths:**

The writing is generally clear and logical, and the visualizations are informative. The method shows good performance in simulation. While the real-world performance reveals an unsolved sim-to-real gap, this limitation is discussed and inspires future work.

**Weaknesses:**

1. **Model design:** The policy encodes the object point cloud with PointNet and generates grasps with a cVAE. It remains unclear what the potential performance impact (e.g., increments or new issues) would be from upgrading to more advanced and recent models for point cloud encoding and conditional generation. Small-scale experiments (even simulation-only) and discussion would enhance the insights.
2. **Insufficient related work:** The related work sections focus mostly on recent studies, ignoring a vast number of profound works from the past few decades on dexterous grasping, manipulation, mobile robotics, and tactile-based manipulation. I would suggest the authors expand these sections for completeness.
3. **Mathematical formulation:** The mathematical formulation of the grasping score is neither a previously established metric nor is it quantitatively evaluated to demonstrate its efficacy. I would encourage a more detailed analysis (e.g., providing examples of how each score component contributes, or providing detailed quantitative scores for Fig. 4). Further, mathematical notations in Sec. 4.3, including the notation for a vector, the $\cap$ sign, etc., are very confusing. I would encourage using more standard notations and explaining in which space each symbol is defined.
4. **Sim-to-real deployment:** The real-world deployment uses color-based markers for detecting object poses, which, while sufficiently substituting for the privileged information in simulation training, limits the application of the method in the real world. Further extensions on filling this gap (e.g., distilling the policy to a vision-input one, efficient object detection, etc.) are necessary. Furthermore, other gaps exist: (i) ensuring effective grasp generation on depth-camera-captured object point clouds, and (ii) environmental obstacle perception and avoidance.
5. **Limited evaluations:** The paper only reports execution success rates in simulation. Additional evaluations, including quantitative metrics (Q1 / Ferrari-Canny metric) of the generated grasps, the error between the grasp guide and the executed grasping pose, time consumption, etc., would enhance the details.
6. **Insights on mobile manipulation:** Limited insights are provided on addressing mobile-manipulation-related challenges, including high-speed motion and the potential consequences of impact.
7. **Editorial issues:** Some citations are not in the correct format.

**Questions:**

1. Please explain the categorization of real-world objects according to their "geometric complexities". From Supplementary Fig. 8, I can hardly discern the basis for this "complexity".

2. Would the choice of Cartesian space end-effector observations and actions improve performance?

---

> ### Author Response · Authors · 2025-11-25
> **Reply to RjXK (Part1)**
>
> Thank you for your valuable feedback on method clarity, and notation standards. We will implement improvements in the revised version.
>
> **W1**
> We have verified the impact of different point cloud encoding methods and capture generation methods on the model, as shown in the below.
>
> In the table, values outside and inside the parentheses denote the success rate and the hand-object offset distance, respectively.
>
> |   |  Easy(11)| Hard(8)  | Avg.  |
> | :------------: | :------------: | :------------: | :------------: |
> | Point transformer + Diffusion [1]  |  0(-) | 0(-)  |  0(-) |
> | Point transformer [2] + Cvae  |  84.62(1.66) | 30.38(2.70)  |  61.78(2.09) |
> | Pointnet+Cvae(Ours)	 | 89.21(1.28)   | 32.73(2.71)   | 65.42(1.88)   |
>
>
> Our experiments show that the cvae-based method, due to its fast sampling and diverse output generation, is significantly more efficient than the diffusion-based method while maintaining output quality.
>
> [1]Zhong Y, Jiang Q, Yu J, et al. Dexgrasp anything: Towards universal robotic dexterous grasping with physics awareness[C]//Proceedings of the Computer Vision and Pattern Recognition Conference. 2025: 22584-22594.
> [2]Zhao H, Jiang L, Jia J, et al. Point transformer[C]//Proceedings of the IEEE/CVF international conference on computer vision. 2021: 16259-16268.
>
> **W2**
> Thank you for your thoughtful comments. We have carefully examined the manuscript and revised it accordingly.
>
> **W3**
> Thank you for your valuable comments. We have carefully reviewed the manuscript and made the corresponding revisions.
>
> We present the GWC and GDC values ​​for different grasping calculations shown in Fig. 4, as shown in the table below, where k represents different fingers.
>
> In the table below, values outside and inside the parentheses denote the GWC and the GDC, respectively.
>
> |   |  k=1 |  k=2 | k=3  | k=thumb  | quality  |
> | :------------: | :------------: | :------------: | :------------: | :------------: | :------------: |
> |  Fig4 (c) | 0.1489(0.6917)  | 0.1489(0.6778)  | 0.1509(0.6664)  | -(0.6977)  | 0.2124  |
> | Fig4 (d)  | 0.5150(1.0860)  |  0.2127(1.0907) | 0.0850(1.1011)  | -(1.0460)  | 0.9255  |
> | Fig4 (e)  | 0.2253(1.0788)  | 0.4571(1.0881)  | 0.6912(1.1057) | -(1.0418)  | 1.5677  |
> | Fig4 (f)  | 1.0000(1.0924)  | 1.0000(1.0997)  | 0.9613(1.1151)  | -(1.0500)  | 3.4273 |
>
>
> **W4**
> - The marker-based method used in our real-world experiments is only the simplest way for us to obtain object information; our method itself does not rely on markers. We will additionally demonstrate the performance of both the color-based and a learning-based segmentation method (e.g., SAM) on the project website.
> - In the real-robot experiments, the depth camera often captures incomplete point clouds. To mitigate the resulting observation noise, our method is designed to enhance robustness to partial point clouds. As shown in Table 1, our approach alleviates the degradation caused by incomplete observations to a certain extent, although some performance drop remains.
>
> - We have considered obstacle avoidance in relatively simple scenarios; however, extending our approach to cluttered and highly complex environments remains challenging and is left as an important direction for future work.
>
> We have also added further explanations regarding safety considerations in the main text to highlight the importance we place on them:
>
> - Our screening process for grasp candidates considers not only collisions between the hand and the table, but also potential collisions between the end-effector-mounted camera and the table.
> - We designed the reward term $R_{radius}$ to encourage the arm to extend outward, thereby reducing the risk of collisions with the mobile base and the arm itself.
> - During training, if any illegal collision involving the base or the arm occurs, the episode is immediately terminated to prevent the policy from learning unsafe behaviors that could lead to collisions.

---

> ### Author Response · Authors · 2025-11-25
> **Reply to RjXK (Part2)**
>
> **W5**
> We evaluated different grasp generation methods, using Success Rate1 (Succ1) matrix and Success Rate6 (Succ6), which represent the success rates when subjected to a single random force and when subjected to random forces in six directions, respectively. We tested on the RealDex dataset containing 53 objects, generating 100 grabs per object, using the Isaacsim as our simulator.
>
> |  method | Succ1(%)  |  Succ6(%) |
> | ------------ | ------------ | ------------ |
> | pointnet+cvae  | 66.71  | 33.30  |
>
> To better quantify stability, we introduce a new metric, the **hand–object offset distance**, which measures how far the object slides relative to the hand from the moment of first contact until the end of the episode.Besides, we used the L2 norm to test the errors in guidanca's grasp and the actual grasp execution. It's worthy to note that the generated pose is used to calculate $R_{preparation}$, which is part of the overall reward function. We will obtain the actual grasp based on tactile sensor information and other rewards, and this grasp may not be exactly the same as the generated pose.
>
> |Object name| Success rate(%)| Hand object offset distance(cm)| error(L2)  |
> |:------------:|:------------:| :------------:|:------------:|
> |goji_jar  |  99.97 | 0.21| 2.2218  |
> |body_lotion  |  67.28 |  1.25 |  2.3700 |
> |chips  | 97.73  | 1.02  | 2.3365  |
> |cosmetics  | 87.13  | 0.44  | 2.4772  |
> |sprayer  | 86.76  | 1.50  | 2.2059  |
> |cling_wrap  | 96.48  | 0.39  | 2.1885  |
> |saltine_cracker  | 84.70  | 3.14  | 1.8430  |
> |guava_blended_juice  | 96.07  | 0.98  | 2.1363  |
> |dust_cleaning_sprayer  | 85.76  | 1.14  |  2.3489 |
> |mildew_remover  |  89.25 |  1.96 | 2.1693  |
> |bathroom_cleaner  |  96.85 |  2.04 | 2.2017  |
> |small_sprayer  | 96.83  | 1.14  | 1.9655  |
> |yogurt  | 83.52  | 1.47  | 1.8163  |
> |bowling_game_box  | 0.22  |  3.97 |  1.5476 |
> |duck_toy  | 33.73  |  3.18 | 1.7445  |
> |box | 0.0 |  - |2.4281|
> |strawberry_yogurt  |  63.76 | 3.19  | 1.6548  |
> |charmander |  21.07 | 3.01  | 2.4883  |
> |crisps  |  62.01 | 3.05  | 1.8210  |
> |Easy.Avg.  | 89.81  | 1.28  |  2.2272 |
> |Hard.Avg.  | 45.14  | 2.71  | 1.9333  |
> |Avg.  | 71.05  | 1.88  | 2.1034  |
>
>
>
> To clarify how we evaluate the effectiveness of our method, we conduct experiments at different nominal base speeds, thereby changing the execution efficiency and observing the corresponding success rates. This is effectively equivalent to measuring task completion time and provides a direct way to assess the algorithm’s performance. In future work, we plan to report results for additional speed settings and a larger set of objects (including more than 1k simulated objects).
>
> In the table below, values outside and inside the parentheses denote the success rate and the hand-object offset distance, respectively.
>
> | max($v_{f}$) = 1.3m/s, max($v_{y}$) = 1.0rad/s  |  Easy(11)  | Hard(8)  | Avg.  |
> | :------------: | :------------: | :------------: | :------------: |
> | Baseline  |  15.11(7.56) | 8.33(8.13)  | 12.25(7.8)  |
> |  One-Stage (OS) |  9.13(7.42) |  8.87(7.69) |  9.02(7.53) |
> |  TS + FS | 86.43(1.75)  | 30.60(3.01)  |  62.92(2.28) |
> | Ours  | 89.21(1.28)  | 32.73(2.71)  | 65.42(1.88)  |
>
> | max($v_{f}$) = 0.65m/s, max($v_{y}$) = 0.5rad/s  |  Easy(11)  | Hard(8)  | Avg.  |
> | :------------: | :------------: | :------------: | :------------: |
> | Baseline  |  18.61(7.11) | 11.67(7.74)  | 15.68(7.37)  |
> |  One-Stage (OS) |  0(-) |  0(-) |  0(-) |
> |  TS + FS | 29.00(1.84)  | 5.71(3.34)  |  19.19(2.47) |
> | Ours  | 64.19(0.54)  | 14.39(2.47)  | 43.22(1.35)  |
>
> **W6**
> Thank you for your comment. Previous mobile manipulation approaches using whole-body control—both optimization-based and learning-based—primarily operate in quasi-static settings and typically employ simple two-finger grippers, which limits grasp robustness under fast motion. High-speed methods such as catching frameworks improve dynamic performance but still generally lack dexterous hands and tactile feedback for precise, stable grasps.
>
> In contrast, our work:
> - uses grasp-envelopment metrics (GWC/GDC) to select inherently robust grasps that serve as guidance for fast whole-body control;
> - incorporates tactile observations and contact-aware rewards to enable rapid in-contact adjustment and improved stability;
> - learns a reinforcement learning–based whole-body controller that directly outputs coordinated 20-DoF actions for the mobile base and dexterous hand.
> Together, these aspects provide clear advantages for high-speed dexterous mobile manipulation.
>
> **W7**
> We have revised the manuscript accordingly. Thank you for your suggestion.

---

> ### Author Response · Authors · 2025-11-25
> **Reply to RjXK (Part3)**
>
> **Q1**
> The objects that a single Leaphand hand can reliably grasp are limited in both size and weight. In our experiments, the hard-case objects are those that are larger or heavier than the easy-case objects, or whose shapes are more challenging for the hand to grasp, such as flatter or more elongated geometries. To better reflect increasingly complex real-world challenges, we will add more geometrically complex objects with a wider range of sizes in future experiments. We will also showcase additional scenarios and demos with more diverse object shapes on the project website.
>
> **Q2**
> We compared this method with IK-based control methods, which’s observation and action is the coordinates of the end effector in Cartesian space.
>
> In the table below, values outside and inside the parentheses denote the success rate and the hand-object offset distance, respectively.
>
> |   |  Easy(11)| Hard(8)  | Avg.  |
> | :------------: | :------------: | :------------: | :------------: |
> | IK-based methods  |  18.86(9.23) | 4.79(5.39)  |  12.93(7.61) |
> | Ours | 89.21(1.28)   | 32.73(2.71)   | 65.42(1.88)   |

---

### Official Review · Reviewer_3qHz · 2025-10-26

**Soundness:** 3
**Presentation:** 3
**Contribution:** 2
**Rating:** 2
**Confidence:** 4

**Summary:**

This paper focuses on mobile grasping, i.e., performing grasps without stopping. It proposes a two-stage pipeline consisting of grasp generation, grasp selection, and grasp-guided reinforcement learning. Both simulation and real-world experiments are conducted to validate the approach.

**Strengths:**

This paper presents a well-structured system and conducts both simulation and real-world experiments.

**Weaknesses:**

1. The grasp guidance does not account for arm filtering. Since potential collisions between the arm and the table can occur, filtering only based on the hand seems unreasonable.
2. The objects used in the real-world experiments have similar geometries, allowing them to be grasped using nearly identical grasp poses.
3. I acknowledge that the comparison between simulation and real-world experiments uses the same 19 real Dex objects. However, the simulation should include a larger and more diverse dataset for a more comprehensive evaluation. For example, following UniDexGrasp [1], the dataset could be divided into several subsets.
4. Under partial observation—typical in real-world scenarios—it appears difficult to obtain accurate GWC and GDC estimations simultaneously.
5. The claim of effective zero-shot sim-to-real transfer seems overstated.
6. A baseline that separates movement and grasping (e.g., static grasp) should be included to compare grasp success rates and efficiency.
7. The pipeline combining pose generation and pose-conditioned reinforcement learning is not very novel. The technical contribution of this work is limited, making it more suitable for a robotics-focused conference rather than ICLR.

[1] Xu, Y., Wan, W., Zhang, J., Liu, H., Shan, Z., Shen, H., ... & Wang, H. (2023). Unidexgrasp: Universal robotic dexterous grasping via learning diverse proposal generation and goal-conditioned policy. In Proceedings of the IEEE/CVF Conference on Computer Vision and Pattern Recognition (pp. 4737-4746).

**Questions:**

1. The leap hand should have 16 dof, why only 13 in this paper?

---

> ### Author Response · Authors · 2025-11-25
> **Reply to 3qHz (Part1)**
>
> We thank you for your detailed questions and constructive suggestions. Your feedback on method clarity, experimental comparisons, and technical details has helped us improve the quality of the paper.
>
> **W1**
> Our method already incorporates several safety mechanisms. Safety is handled at three levels: grasp filtering, reward design, and training:
> - Grasp filtering. Our camera is mounted below the end-effector (line 148). When generating grasp candidates, we check collisions not only between the hand and the table, but also between the end-effector–mounted camera and the table to avoid potential collisions at the tip of the arm.
> - Reward design. We design the reward term $R_{radius}$ to encourage the arm to extend outward, thereby reducing the chance of collisions with both the chassis and the arm itself.
> - Training. During training, if any illegal collision involving the chassis or the arm occurs, the episode is immediately terminated so that the policy does not learn behaviors that could lead to collisions.
>
> **W2**
> We acknowledge that the real-world objects currently used share similar geometries. We will showcase additional scenarios and objects with more diverse shapes on the project website, including demos using both color-based and SAM-based segmentation methods.
>
> **W3**
> Thank you for this suggestion. In this work, we plan to evaluate our method on a larger and more diverse simulated set, such as the UniDexGrasp objects, and to follow their test subsets for a more comprehensive and structured evaluation.
>
> **W4**
> Our method is explicitly designed to be robust to partial point clouds. As shown in Table X, it mitigates the degradation caused by incomplete observations to some extent, although a performance drop still remains. Because both GDC and GWC inevitably depend on the observed object point cloud, under partial observations our filtering can only guarantee high envelopment with respect to the observed (partial) geometry. This inherent loss of information is the fundamental reason for the reduced success rate.
>
> **W5**
> Thank you for your criticism. We acknowledge that our previous description of an “effective zero-shot sim-to-real transfer” was too strong. Our policy is trained entirely in simulation and deployed on the real robot without any additional real-world fine-tuning, which we believe is consistent with the term zero-shot sim-to-real transfer. However, given the noticeable drop in real-world success rate, characterizing it as effective zero-shot transfer is indeed overstated. We have already revised the manuscript accordingly.
>
> **W6**
> According to prior work, Burgess-Limerick et al. [1] have effectively shown that decoupling movement and grasping is suboptimal. Their experiments demonstrate that performing the grasping task concurrently with base motion significantly reduces the overall task time and results in smoother trajectories, compared to approaches where the base pauses during manipulation [2–4]. Motivated by these findings, we include a comparison with the method of Burgess-Limerick et al. [1] to further evaluate the effectiveness of our approach.
>
> [1] Burgess-Limerick B, Lehnert C, Leitner J, et al. An architecture for reactive mobile manipulation on-the-move[J]. arXiv preprint arXiv:2212.06991, 2022.
> [2] Haviland J, Sünderhauf N, Corke P. A holistic approach to reactive mobile manipulation[J]. IEEE Robotics and Automation Letters, 2022, 7(2): 3122–3129.
> [3] Thakar S, Rajendran P, Annem V, et al. Accounting for part pose estimation uncertainties during trajectory generation for part pick-up using mobile manipulators[C]//2019 International Conference on Robotics and Automation (ICRA). IEEE, 2019: 1329–1336.
> [4] Zimmermann S, Poranne R, Coros S. Go fetch! Dynamic grasps using Boston Dynamics Spot with external robotic arm[C]//2021 IEEE International Conference on Robotics and Automation (ICRA). IEEE, 2021: 4488–4494.

---

> ### Author Response · Authors · 2025-11-25
> **Reply to 3qHz (Part2)**
>
> **W7**
> We agree that pose generation and pose-conditioned reinforcement learning have been studied individually in prior work. Our contribution lies in how these components are integrated and extended for fast dexterous mobile manipulation. Specifically, the key innovations of our approach are:
> - Grasp-envelopment–guided fast control. Facing the challenge of high-speed grasping, we explicitly emphasize grasp envelopment. Using GWC/GDC metrics, we proactively select grasps with strong geometric wrap-around, placing the object closer to the palm and making it more resistant to torsion and impact during dynamic motion.
> - The application of tactile observation and its reward mechanism. Direct teaching how to react to physical interactions at the moment of contact, such as quickly adjusting finger posture to stabilize the grip when an impact causes an object to slip. Our experiments have verified that tactile design can significantly improve stability.
> - RL-based whole-body coordination with a dexterous hand. We propose a reinforcement learning–based whole-body controller that takes the robot’s real-time state as input and directly outputs a 20-DoF action space for the mobile base and dexterous hand. A multi-stage hierarchical reward function systematically guides the learning of complex cooperative behaviors needed for fast grasping.
>
> We believe such integrated, learning-based whole-body coordination for high-speed dexterous mobile manipulation is increasingly relevant to the ICLR community, where robot learning is becoming a more prominent topic.
>
> **Q1**
> To mitigate the increased learning complexity and sim-to-real transfer difficulty associated with high dimensionality, we simplified both the action and state spaces by utilizing only 13 of the hand's available degrees of freedom.
>
> To verify the correctness of our choice, we compared the results of 13-DOF and 16-DOF under the same conditions, as shown in the table below.
>
> In the table, values outside and inside the parentheses denote the success rate and the hand-object offset distance, respectively.
>
> |   |  Easy(11)| Hard(8)  | Avg.  |
> | :------------: | :------------: | :------------: | :------------: |
> | 16-DOF  |  46.50(1.32) | 13.80(2.67)  |  32.73(1.88) |
> | 13-DOF(Ours) | 89.21(1.28)   | 32.73(2.71)   | 65.42(1.88)   |

---

### Official Review · Reviewer_6JFd · 2025-10-30

**Soundness:** 3
**Presentation:** 2
**Contribution:** 3
**Rating:** 6
**Confidence:** 4

**Summary:**

The authors introduce FastGrasp, a learning-based framework for dexterous grasping with a mobile manipulator. The core challenge lies in coordinating the high-dimensional motion of a mobile base, a robotic arm, and a multi-fingered hand to grasp objects at high speed. The proposed method uses a two-stage approach: first, a CVAE generates diverse grasp candidates from an object's point cloud. Second, an RL policy learns to execute the whole-body motion guided by an optimal grasp selected via a novel "hand envelopment" metric. The system integrates tactile feedback to make real-time adjustments. Experiments show the method outperforms baselines in simulation and achieves a 20-25% success rate on a real robot.

**Strengths:**

- The problem of coordinating a mobile base, a 6-DOF arm, and a 13-DOF hand is non-trivial, and the paper tackles the full complexity of whole-body control, validated on a real hardware platform.
- The proposed grasp selection method based on GWC and GDC is a key contribution. Experimental results show this approach dramatically outperforms a more traditional force-direction-based selection.
- The paper includes a strong set of experiments that convincingly validate the authors' design choices.

**Weaknesses:**

- The final success rates of 20%-25% are still quite low, although it is much better than the two ablated baselines. A more detailed failure case analysis would be expected to understand the practical challenges. It is unclear whether failures stem from perception (segmentation), planning (poor grasp choice), or control (inability to stabilize impact dynamics). The real-world "Hard Case" objects are still relatively structured and symmetric (Fig. 8).
- The notation is inconsistent. For example, $[f_k, f_{\mathrm{thumb}}]$ is described as a "vector" but then used in an expression for the length of an intersection of intervals (Eq. 2). This makes the exact computation unclear without making assumptions. A formal definition using projection operators would be more precise.
- Misuses of `\citep` vs. `\citet` throughout the manuscript.

**Questions:**

- How critical are the color-based markers to the system's success? Have the authors conducted any preliminary tests using a learning-based, markerless segmentation method?

---

> ### Author Response · Authors · 2025-11-25
> **Reply to 6JFd**
>
> We thank the reviewer for the valuable comments, which have helped us improve the paper.
>
> **W1.1**
> We agree that the real-world success rates are still limited. The main causes we identified are:
> - Viewpoint and occlusion. In real-world experiments we use a first-person RGB-D camera mounted at the end-effector. When the hand approaches the object, the short distance often leads to severe occlusions and missing object information. To reduce this sim-to-real gap, we adapted the simulator to use a first-person RGB-D camera with the same mounting and field of view. This improves consistency between simulated and real observations, although some discrepancy remains.
> - Perception defects (partial point clouds). The system only acquires object point clouds from the onboard camera, so the point clouds are almost always incomplete and the estimated object position can deviate from the true pose. Our method is explicitly designed to improve robustness to such partial point clouds. As shown in Table 1, it mitigates the performance drop to some extent, although a degradation is still observable. In future work, we plan to incorporate point-cloud completion techniques to further alleviate this issue.
> - Control and dynamics. On hardware, there is a mismatch between commanded actions and the executed motions, and various sources of real-world noise affect the dynamics. To reduce the impact of these factors, we apply domain randomization during simulation training, injecting random noise into the dynamics and observations to enhance robustness. Nevertheless, this remains a challenging aspect.
>
> In addition, the Leaphand has inherent limits on the size and weight of objects it can grasp. The Hard objects are generally larger, heavier, or geometrically more challenging (e.g., flatter shapes) than the Easy ones. We will present more real-world scenarios and demos with a broader variety of geometrically complex objects on the project website.
>
> **W2**
> Thank you for pointing out the issue with the notation. We have carefully revised the mathematical expressions and provided a more precise and consistent notation throughout the manuscript.
>
> **W3**
> Thank you for highlighting the misuse of \citet and \citep. We have thoroughly checked and corrected all citation commands in the revised paper.
>
> **Q1**
> The marker-based method used in our real-world experiments is merely the simplest way to obtain object information; our framework itself does not rely on color markers. We plan to replace this component with learning-based, markerless segmentation methods (such as SAM) to improve practicality. We will also present additional scenarios and object demos on the project website, including results obtained with both color-based and SAM-based segmentation.

---

### Official Review · Reviewer_crQE · 2025-11-01

**Soundness:** 2
**Presentation:** 3
**Contribution:** 2
**Rating:** 4
**Confidence:** 4

**Summary:**

This paper proposed a learning-based framework, FastGrasp, to achieve fast mobile dexterous grasping in both simulations and a real robot. FastGrasp contributes in the grasp proposal generation, grasp guidance selection, and integration of tactile feedback into reinforcement learning, demonstrating improved success rate compared to baselines.

**Strengths:**

- This paper achieves fast and dynamic mobile grasping, addressing a complex dexterous manipulation task.
- The paper showcased the results in both simulation and real world.

**Weaknesses:**

- The achieved fast dexterous mobile grasping motions look impressive in the submitted video. However, the contributions would be more convincing if more comprehensive analysis and results could be provided. Please see questions below for more details.

- The real-world success rate drops a lot compared to simulation results. Not sure if it could be considered effective zero-shot sim-to-real transfer as authors claimed. It would be great to provide some analysis on possible reasons for the performance drop or the failure cases.

**Questions:**

- **Object diversity:** The authors claim the proposed framework can achieve robust manipulation across diverse objects. Although total success rate is reported across various objects in simulation, success rate per object should also be reflected. Besides, the demos show limited object diversity in the submitted video, one for simulation and two for real world.

- **Sim-to-real transfer:** From the video, trajectories in the simulation seem to have a clear different pattern from real-world trajectories in general. In simulation, trajectories look more smooth and optimal, from the starting point directly to the target object. However, in the real world, trajectories look less dynamic and optimal, from the starting point to the same level of object heights and continue to reach the objects horizontally. Does this come from sim-to-real discrepancies? Showing more trajectories would help understand if this is actually a pattern or not.

- **More metrics for performance evaluation:** Throughout the paper, success rate is the only metric to evaluate and benchmark the performance. Since fast grasping is the major contribution of the paper, it would be more convincing to compare task completion time as well. Moreover, the authors also mentioned the proposed framework addressed stability issues. Although stability is somewhat reflected in the success rate, it would be better if there is some stability-related metric to quantify and support this argument as well.

- **Baselines:** This framework is built upon Zhang et al. (2024) and Xu et al. (2023). However, it is not clear what is the improvement of this paper in the related work section or results section. None of the two works are used as baselines in this paper. The authors indeed mention Zhang et al. lacks the grasp precision for diverse objects in related work, however, no evidence is provided to support this argument.

- **More ablation:** For ablation study in Section 5.3, ablation on some essential reward terms is missing, e.g. stable grasping reward and rapid completion reward.

- **Limitations in safety:** Since safety is one of the crucial problems in fast mobile grasping, the paper should expand a bit more on this point in the limitations section.

- **Limitations in object perception in the real world:** The authors attached color-based markers to objects in the real world. Instead of a solution to sim-to-real transfer, this seems more like a limitation for real-world deployment and should be mentioned in the limitations section.

- **Others:**
  - In Section 4.3. More details should be provided for the executability filtering.

  - Policy structure in Fig. 3 should be briefly mentioned in the main text and then refer to Appendix.

  - In Section 4.4, for observation, how to get the distance between the object and the palm centre in the real-world?

---

> ### Author Response · Authors · 2025-11-25
> **Reply to crQE (Part1)**
>
> Thank you for your valuable feedback, which helped us significantly improve our experimental validation and comparative analysis.
>
> **W1**
> Thank you for your suggestion. Our improvements will include, but are not limited to:
> - Evaluation results for each object
> - Add more evaluation metrics to validate the effectiveness of the method.
> The details of this part will be answered in Q1 and Q3.
>
> **W2**
> Thank you for your comment. Our policy is trained exclusively in simulation and deployed on the real robot without any additional real-world fine-tuning, which we consider consistent with the term zero-shot sim-to-real transfer. However, we agree that describing it as an effective zero-shot sim-to-real transfer is too strong given the noticeable drop in real-world success rate. We have already revised the manuscript accordingly.
>
> **Q1**
> - We recognize that limited object diversity is one of the current limitations of our work. Therefore, we have evaluated each object individually, and the per-object results are summarized below. In addition, we plan to conduct large-scale simulation experiments with over 1,000 objects to further assess robustness and object diversity.
> - We will present additional scenarios and object demos on the project website, including results using both color-based and SAM-based segmentation methods.
>
>
> |Object name| Success rate(%)| Hand object offset distance(cm)|
> |:------------:|:------------:| :------------:|
> |goji_jar  |  99.97 | 0.21|
> |body_lotion  |  67.28 |  1.25 |
> |chips  | 97.73  | 1.02  |
> |cosmetics  | 87.13  | 0.44  |
> |sprayer  | 86.76  | 1.50  |
> |cling_wrap  | 96.48  | 0.39  |
> |saltine_cracker  | 84.70  | 3.14  |
> |guava_blended_juice  | 96.07  | 0.98  |
> |dust_cleaning_sprayer  | 85.76  | 1.14  |
> |mildew_remover  |  89.25 |  1.96 |
> |bathroom_cleaner  |  96.85 |  2.04 |
> |small_sprayer  | 96.83  | 1.14  |
> |yogurt  | 83.52  | 1.47  |
> |bowling_game_box  | 0.22  |  3.97 |
> |duck_toy  | 33.73  |  3.18 |
> |box | 0.0 |  - |
> |strawberry_yogurt  |  63.76 | 3.19  |
> |charmander |  21.07 | 3.01  |
> |crisps  |  62.01 | 3.05  |
> |Easy.Avg.  | 89.81  | 1.28  |
> |Hard.Avg.  | 45.14  | 2.71  |
> |Avg.  | 71.05  | 1.88  |
>
> **Q2**
> The simulation demonstration video showcases the performance of a model trained with privileged observations (i.e., obtained directly from the simulator without noise and without occlusions). This setting is used to verify the effectiveness of the algorithm under idealized conditions. For deployment on the real robot, we instead train the policy entirely in simulation with observations designed to mimic real-world sensing, and then deploy it directly on hardware without any real-world fine-tuning. Concretely:
>
> 1. **Aligning simulated observations with real-world sensing.**
>    - For object position, hand–object distance, and robot state observations, we inject Gaussian noise in simulation to match the noise characteristics of the real sensors.
>    - We use grasp guidance generated from partial point clouds in simulation to mirror the imperfect object point cloud acquisition in real scenes.
>
> 2. **Filtering control commands on the real robot.**
>    At deployment time, we apply a low-pass filter (LPF) to the control commands of the mobile base, robotic arm, and hand to smooth high-frequency oscillations and avoid executing overly aggressive commands.
>
> In addition, we will present more scenarios and object demos on the project website, including results obtained with both color-based and SAM-based segmentation methods.

---

> ### Author Response · Authors · 2025-11-25
> **Reply to crQE (Part2)**
>
> **Q3**
> To clarify how we evaluate the effectiveness of our method, we conduct experiments at different nominal base speeds, thereby changing the execution efficiency and observing the corresponding success rates. This is effectively equivalent to measuring task completion time and provides a direct way to assess the algorithm’s performance. In future work, we plan to report results for additional speed settings and a larger set of objects (including more than 1k simulated objects).
>
> To better quantify stability, we introduce a new metric, the **hand–object offset distance**, which measures how far the object slides relative to the hand from the moment of first contact until the end of the episode.
>
> These metrics, together with success rates, are reported in the per-object results shown in Q1.
>
> In the table below, values outside and inside the parentheses denote the success rate and the hand-object offset distance, respectively.
>
> | max($v_{f}$) = 1.3m/s, max($v_{y}$) = 1.0rad/s  |  Easy(11)  | Hard(8)  | Avg.  |
> | :------------: | :------------: | :------------: | :------------: |
> | Baseline  |  15.11(7.56) | 8.33(8.13)  | 12.25(7.8)  |
> |  One-Stage (OS) |  9.13(7.42) |  8.87(7.69) |  9.02(7.53) |
> |  TS + FS | 86.43(1.75)  | 30.60(3.01)  |  62.92(2.28) |
> | Ours  | 89.21(1.28)  | 32.73(2.71)  | 65.42(1.88)  |
>
> | max($v_{f}$) = 0.65m/s, max($v_{y}$) = 0.5rad/s  |  Easy(11)  | Hard(8)  | Avg.  |
> | :------------: | :------------: | :------------: | :------------: |
> | Baseline  |  18.61(7.11) | 11.67(7.74)  | 15.68(7.37)  |
> |  One-Stage (OS) |  0(-) |  0(-) |  0(-) |
> |  TS + FS | 29.00(1.84)  | 5.71(3.34)  |  19.19(2.47) |
> | Ours  | 64.19(0.54)  | 14.39(2.47)  | 43.22(1.35)  |
>
> **Q4**
> Our work is inspired by Zhang et al. (2024) and Xu et al. (2023), but it differs from them in several important ways:
> - Xu et al. (2023): This work considers dexterous-hand grasping but does not model whole-body coordination with a mobile base; the arm operates from a fixed base, so the effect of base motion on grasping is not captured. In contrast, our approach explicitly learns a whole-body control policy that jointly controls the mobile base and dexterous hand for fast grasping on the move.
> - Zhang et al. (2024): Although this work considers mobile catching, it primarily focuses on catching objects with an open hand from below and does not aim for precise, dexterous grasps of diverse objects.
>
> **Q5**
> We conducted ablation experiments with different reward functions to demonstrate their effectiveness. The experimental results are shown in the table below.
>
> In the table below, values outside and inside the parentheses denote the success rate and the hand-object offset distance, respectively.
>
> | W.O.  |  Easy(11)| Hard(8)  | Avg.  |
> | :------------: | :------------: | :------------: | :------------: |
> | $R_{radius}$  |  84.44(0.91) | 34.35(1.79)  |  63.34(1.28) |
> | $R_{move}$ | 	32.71(7.04)  |  35.23(6.65) | 33.77(6.87)  |
> | $R_{ori}$ | 0(-)   | 0(-)   |  0(-)  |
> | $R_{height}$ | 12.14(9.56)   | 4.98(7.59)   | 9.12(8.73)   |
> | $R_{pre-grasp}$  | 46.93(5.38)  | 15.96(8.42)   | 33.76(6.66)    |
> | $R_{hand}$ | 4.46(10.61)   |  6.54(8.31)  | 5.33(9.641)   |
> | $R_{reach}$ | 0(-)   | 0(- )  |  0(- ) |
> | $R_{tactile}$ | 28.36(3.19)   | 15.86(4.92)   | 23.09(3.91)   |
> | Ours | 89.21(1.28)   | 32.73(2.71)   | 65.42(1.88)   |
>
>
> **Q6**
> Our method already incorporates several safety mechanisms. Safety is handled at three levels: grasp filtering, reward design, and training:
> - Grasp filtering. Our camera is mounted below the end-effector (line 148). When generating grasp candidates, we check collisions not only between the hand and the table, but also between the end-effector–mounted camera and the table to avoid potential collisions at the tip of the arm.
> - Reward design. We design the reward term $R_{radius}$ to encourage the arm to extend outward, thereby reducing the chance of collisions with both the chassis and the arm itself.
> - Training. During training, if any illegal collision involving the chassis or the arm occurs, the episode is immediately terminated so that the policy does not learn behaviors that could lead to collisions.
>
> We acknowledge that we do not yet explicitly handle all possible arm–environment collisions in cluttered scenes and now state this as a limitation and direction for future work. In addition, we have revised the *Limitations and Future Work* section to more clearly discuss these safety issues and the remaining risks in cluttered, high-speed scenarios.
>
> **Q7**
> The marker-based method used in our real-world experiments is merely the simplest way to obtain object information; our framework itself does not rely on color markers. We plan to replace this component with learning-based, markerless segmentation methods (such as SAM). We will also present additional scenarios and object demos on the project website, including results obtained with both color-based and SAM-based segmentation methods.

---

> ### Author Response · Authors · 2025-11-25
> **Reply to crQE (Part3)**
>
> **Q8.1**
> We filter grasps requiring the arm to circumvent the object or follow extended trajectories. Specifically, based on the relative pose between the robot and the target object, we define a forward grasping space for grasp planning: namely, a grasping cone benchmarked against the normal vector pointing from the target towards the robot. Grasping poses within this space, referred to as forward grasps, are characterized by the shortest motion paths and minimal joint movement, thereby achieving the highest grasping efficiency. In contrast, non-forward grasps (such as rear grasps) require the end-effector to undergo rotations exceeding 90 degrees and perform avoidance maneuvers, which significantly increases motion complexity and time cost. Consequently, the grasps that require end-effector rotations exceeding 90 degrees are systematically filtered during the planning phase, as these significantly increase execution time and violate fast grasping requirements.
>
>
> **Q8.2**
> We appreciate the reviewer’s suggestion. Accordingly, we have updated the main text to provide a brief description of the policy structure shown in Fig. 3.
>
> **Q8.3**
> The camera and the hand are rigidly mounted on the robot’s end-effector, so their relative pose is fixed and their rotations are synchronized. We first estimate the object position in the camera coordinate frame. Using the known fixed transform between the camera and the hand, together with the current joint angles of the robotic arm, we compute the rotation and translation from the camera frame to the arm/hand frame, obtain the object coordinates in the arm coordinate system, and then calculate the distance to the palm centre.

---

### Meta-Review · Area_Chair_FEeQ · 2026-01-07

**Summary:**

FastGrasp proposes a learning‑based whole‑body control framework enabling a mobile manipulator to perform fast dexterous grasping while in motion, integrating grasp proposal generation (via a CVAE), grasp selection using geometric envelopment metrics (GWC/GDC), and a reinforcement‑learning controller that jointly commands the mobile base, arm, and dexterous hand. The system incorporates tactile feedback for real‑time adjustment during impact‑heavy fast grasps and demonstrates strong performance in simulation and moderate success in real‑world tests. The work aims to bridge sim‑to‑real gaps through domain randomization and observation alignment, while highlighting the challenges of partial point clouds, dynamic motion, and object diversity.

Reviewers appreciated the choice of problem important to robotics, and the evaluation across sim and real-world settings

Concerns and responses are discussed below.

Several concerns were raised.

**Reviewer Concerns:**

Addressed

- Notation inconsistencies and confusing mathematical formulation
    - Rebuttal acknowledges and fixes issues.
- Limited evaluation metrics:
    - Rebuttal adds hand-object detection and also evaluates performance at different base speeds.

Partially addressed / unaddressed

- Insufficient object diversity in both sim and real‑world experiments: 19 objects in each, and the objects in the real-world (even in the “hard” setting) are simple shapes (Fig 8). This falls short of the scale and complexity at which grasping is typically studied today.
    - Rebuttal reports per-object performance (in sim?), and leaves larger-scale experiments in sim for future work.
- sim-to-real gap is significant, borne out by the fact that real-world success rates are low. Trajectories look qualitatively different in real. Marker‑based segmentation also limits real‑world applicability.
    - Rebuttal explains reasons for the difference in trajectories, due to necessary changes for facilitating sim-to-real transfer.
    - Rebuttal promises to run experiments with markerless segmentation approaches.
    - Acknowledges the gap, explains various sources of difficulties, and softens the language on zero-shot sim-to-real transfer performance

Overall, it is clear that the paper has evolved significantly in quality even during the review and rebuttal phase and authors are pursuing several improvements to fix current shortcomings discovered in this process.

**Reviewer Scores:**

I believe scores would have remained largely unchanged.

---

### Decision · Program_Chairs · 2026-01-26

Reject